# ST8Sia2 polysialyltransferase protects against infection by *Trypanosoma cruzi*

**Bruno Rafael Barboza[1], Janaina Macedo-da-Silva[1], Lays Adrianne Mendonça Trajano Silva[1], Vinícius de Morais Gomes[1], Deivid Martins Santos[1], Antônio Moreira Marques-Neto[1], Simon Ngao Mule[1], Claudia Blanes Angeli[1], Juliana Borsoi[2], Carolina Borsoi Moraes[3], Cristiane Moutinho-Melo[4,5], Martina Mühlenhoff[6], Walter Colli[7], Suely Kazue Nagashi Marie[8], Lygia da Veiga Pereira[2], Maria Julia Manso Alves[7], Giuseppe Palmisano**[1,9]*

**1** GlycoProteomics Laboratory, Department of Parasitology, Institute of Biomedical Sciences, University of São Paulo, São Paulo, Brazil, **2** Department of Genetics and Evolutionary Biology, Institute of Biosciences, University of São Paulo, São Paulo, Brazil, **3** Department of Clinical and Toxicological Analyses, School of Pharmaceutical Sciences, University of São Paulo, São Paulo, Brazil, **4** Laboratory of Vaccine Development, Department of Microbiology, Institute of Biomedical Sciences, University of São Paulo, São Paulo, Brazil, **5** Laboratory of Immunological and Antitumor Analysis, Department of Antibiotics, Bioscience Center, and Keizo Asami Immunopathology Laboratory, Federal University of Pernambuco, Recife, Brazil, **6** Institute of Clinical Biochemistry, Hannover Medical School, Hannover, Germany, **7** Department of Biochemistry, Institute of Chemistry, University of São Paulo, São Paulo, Brazil, **8** Laboratory of Molecular and Cellular Biology (LIM 15), Department of Neurology, School of Medicine, University of São Paulo, São Paulo, Brazil, **9** School of Natural Sciences, Macquarie University, Sydney, New South Wales, Australia

\* palmisano.gp@usp.br

**Data Availability Statement:** All raw data required to replicate this study are deposited in PRIDE data repository (https://www.ebi.ac.uk/pride/) with the

## Abstract

Glycosylation is one of the most structurally and functionally diverse co- and post-translational modifications in a cell. Addition and removal of glycans, especially to proteins and lipids, characterize this process which has important implications in several biological processes. In mammals, the repeated enzymatic addition of a sialic acid unit to underlying sialic acids (Sia) by polysialyltransferases, including ST8Sia2, leads to the formation of a sugar polymer called polysialic acid (polySia). The functional relevance of polySia has been extensively demonstrated in the nervous system. However, the role of polysialylation in infection is still poorly explored. Previous reports have shown that *Trypanosoma cruzi* (*T. cruzi*), a flagellated parasite that causes Chagas disease (CD), changes host sialylation of glycoproteins. To understand the role of host polySia during *T. cruzi* infection, we used a combination of *in silico* and experimental tools. We observed that *T. cruzi* reduces both the expression of the ST8Sia2 and the polysialylation of target substrates. We also found that chemical and genetic inhibition of host ST8Sia2 increased the parasite load in mammalian cells. We found that modulating host polysialylation may induce oxidative stress, creating a microenvironment that favors *T. cruzi* survival and infection. These findings suggest a novel approach to interfere with parasite infections through modulation of host polysialylation.

dataset identifier PXD053813 and within the manuscript and its supporting information files.

**Funding:** We are grateful for the financial support provided by the São Paulo Research Foundation (FAPESP), grants processes n° 2018/18257-1 (GP), 2018/15549-1 (GP), 2020/04923-0 (GP), 2022/09915-0 (BRB), 2021/00140-3 (JMDS), 2022/00796-9 (LAMTS), 2021/00507-4 (VdMG), 2021/14179-9 (DMS), 2021/14751-4 (SNM), 2020/02988-7 (SKNM), 2023/02096-7 (CMM); by the Conselho Nacional de Desenvolvimento Científico e Tecnológico ("Bolsa de Produtividade") (SKNM, CMM, and GP); by Fundação Faculdade de Medicina (FFM-SKNM); by the Coordenação de Aperfeiçoamento de Pessoal de Nível Superior (AMMN). The funders had no role in study design, data collection and analysis, decision to publish, or preparation of the manuscript.

**Competing interests:** The authors have declared that no competing interests exist.

## Author summary

Glycosylation, the process of adding and removing sugar molecules from proteins, lipids, and small RNAs, plays a crucial role in numerous biological processes. *Trypanosoma cruzi*, the parasite responsible for Chagas disease affecting 6 to 8 million people globally, significantly alters the sialylation of host cell glycoproteins. In mammals, the enzyme ST8Sia2 is involved in forming polysialic acid (polySia), primarily studied in the nervous system, yet its role in infectious diseases remains poorly understood. In our study, we found that *T. cruzi* infection reduces the expression of ST8Sia2 in host cells, leading to decreased levels of polySia. This reduction also affects key proteins such as NCAM1 and SCN5A. Furthermore, experiments involving the removal of polySia, chemical inhibition of ST8Sia2, or genetic silencing of the enzyme resulted in increased intracellular parasites numbers. We propose that altering host polysialylation may induce oxidative stress, thereby favoring *T. cruzi* survival and infection. Our findings suggest that inhibiting ST8Sia2 enhances *T. cruzi* infection, providing new insights into Chagas disease mechanisms and highlighting the important role of host polysialylation.

## Introduction

Glycans are important regulators in different biological processes, with prominent roles in cell adhesion [1], cell activation and signaling [2,3], modulation of immune responses [4,5], cell death [6], and initiation of tumor progression and metastases [7]. The addition and removal of glycans (mono, oligo and polysaccharides) to proteins, lipids [8] and small RNAs [9] is performed by an array of glycosyltransferases and glycosidases, that act in a concerted way to remodel the cellular glycome [10,11]. Glycosylation is one of the most structurally and functionally complex co- and/or post-translational modifications that occur in cells [12,13]. Glycans attached to glycoconjugates, commonly expressed on the surface of cells, exhibit enormous variability and structural complexity, contributing to their respective biological activities [14].

The majority of glycan chains are composed of monosaccharides with five- or six-carbons, with the striking exception of sialic acid (Sia), which belongs to a large family of sugars with a nine-carbon atom backbone and typically found attached to the terminal position of glycans [15]. The glycosidic linkage between a Sia monomer to another underlying Sia residue generates homo-oligo/polymer structures from di- to polysialic acid (polySia) with different $\alpha 2,4/8/9$ intersialyl linkages [16,17]. The biosynthesis of $\alpha 2,8$-linked polySia is catalyzed by ST8Sia polysialyltransferases (polySTs) [18]. In mammals, polysialylation is catalyzed by Golgi-resident polySTs: ST8Sia2 and ST8Sia4 [16,19]. Recently, ST8Sia3 was reported as a polyST capable of autopolysialylation [20]. Structurally, polySia is characterized by repeated Sia monomers with a degree of polymerization ranging from 8 to 400 sialic acid molecules [21]. The presence of polySia is confined to a restricted set of glycoproteins, with the neural cell adhesion molecule 1 (NCAM1) as the best-studied polySia-carrier in mammals [22].

The biological relevance of polySia in mammals has been extensively explored in the brain, highlighting its role in the migration of neuronal precursor cells [23], synaptic neuronal plasticity [19], and axonal outgrowth [24]. The involvement of polySia has also been demonstrated in the development of other organs, including the liver [25], kidney [26], placenta [27], and heart [28]. Beyond the considerable number of studies that have provided important information about the role of polysialylation in the nervous system, the participation of polySTs and polySia in the cardiac system has been less explored. Recently, NCAM and polysialylation have

been shown to play critical roles in the cardiac conduction system [29]. Additionally, in *ST8Sia2* knockout mice, polysialylation deficiency compromised the action potential and the voltage-gated sodium channel functions in atrial cardiomyocytes [30], experimentally evidencing that modulation in polysialylation is sufficient to alter the complex developmental biology and function of cardiomyocytes.

Chagas disease, also known as American trypanosomiasis, is an anthropozoonosis caused by the protozoan *T. cruzi* that affects between 6 and 8 million individuals worldwide, being considered a neglected tropical disease in the world [31,32]. Clinically, CD can be characterized by two main successive phases, the acute and the chronic phases [33]. Among the clinical manifestations of Chagas disease, Chronic Chagas Cardiomyopathy (CCC), presents itself as the most severe manifestation of the disease and affects 1/3 of infected individuals. CCC comprises four main syndromes: *heart failure*, *arrhythmias*, *thromboembolism* and *anginal manifestations* that can simultaneously occur in the same individual [34,35].

In the course of host infection, *T. cruzi* uses a wide variety of mechanisms to escape from host immune surveillance and establish a successful infection, with emphasis on *Trans*-sialidases (TS) [36,37] which catalyze direct Sia transfer between macromolecules, a novel enzymatic activity that was earlier demonstrated [38]. During host cell adhesion and invasion, these enzymes provide an elegant mechanism in which *T. cruzi* specifically captures α2,3-sialic acid from host cells and covers its own surface molecules, mainly mucins, as reviewed in [37,39,40]. Of note, *T. cruzi* parasites are incapable of synthesizing Sia [41,42]. In fact, parasite cells transfer Sias from host cell-surface, the internal surface of the phagolysosome of host cells and from serum sialoglycoproteins to trypanosome cell-surface glycoconjugates [36,43,44].

Previous studies reported that in *T. cruzi* infection, possible disorders of host aberrant glycosylation are due to impairment of glycoprotein sialylation. In 1986, Libby and colleagues [45] demonstrated *in vitro* that *T. cruzi* trypomastigotes modify the surface of rat myocardial and human vascular endothelial cells by desialylation. According to Marin-Neto and colleagues [46], the loss of Sias on the surface of endothelial cells can promote an increase in platelet aggregation and, consequently, microvascular thrombosis. More recently, it was demonstrated that CD43, a sialoglycoprotein expressed at high levels on leucocytes and previously reported as a natural receptor for *T. cruzi* trans-sialidase [47], plays a critical role in the pathogenesis of acute chagasic cardiomyopathy. In this study, a reduced cardiac parasite load in CD43 knockout mice [48] was observed, suggesting the involvement of CD43 in the development of Chagas cardiomyopathy.

Considering the clinical implications of Chagas cardiomyopathy and the current gaps in understanding the involvement of cardiac glycosylation in susceptibility and/or resistance to *T. cruzi* infection, exploring the interplay between glycosylation modulation and possible changes in the pathophysiology of Chagas cardiomyopathy presents an compelling avenue for investigation. This study unveils novel molecular mechanisms exploited by *T. cruzi* to facilitate successful host cell infection. Employing a blend of *in silico* and experimental tools, our investigation assessed the impact of host polysialylation modulation during experimental *T. cruzi* infection. Our results demonstrate that *T. cruzi* infection notably reduces ST8Sia2 expression and substrate polysialylation. Moreover, chemical and genetic inhibition of ST8Sia2 enhances parasite burden within host cells. Additionally, we found that modulation of host polysialylation induces oxidative stress, establishing a conducive microenvironment for *T. cruzi* survival and infection. In summary, our findings underscore the pivotal role of host polysialylation in *T. cruzi* infection dynamics.

## Results

### Modulation of genes involved in host N-glycosylation machinery by *T. cruzi* is accompanied by downregulation of *ST8Sia2* polysialyltransferase

In this study, we first performed an *in-silico* analysis on publicly available transcriptome data obtained from human induced pluripotent stem cell-derived cardiomyocytes (hiPSC-CM) infected with *T. cruzi* Y strain [49] to map the differential expression of glycan-related genes (glycogenes) (**S1 Fig**). A total of 39 glycogenes, including α-glucosidases, fucosyltransferases, galactosyltransferases, glucosyltransferases, mannosidases, mannosyltransferases, n-acetylglu-cosaminyltransferases, sialidases, and sialyltransferases were modulated in infected hiPSC-CM (**S1A Fig**). Among the regulated glycogenes, ST8 alpha-N-acetyl-neuraminide alpha-2,8-sialyl-transferase 2 (*ST8Sia2*) was downregulated in *T. cruzi*-infected hiPSC-CM (**S1B–S1E Fig**), being more pronounced at 24 and 48 hours post-infection (h.p.i.) (**S1D Fig**). A previous study also showed that polysialylation deficiency compromises the functionality of atrial cardiomyocytes in *ST8Sia2* knockout mice [30]. Taken together, these findings strongly suggested that in hiPSC-CM, *T. cruzi* parasite modulates the expression of glycogenes, and manipulates host polysialylation machinery by downregulating ST8Sia2.

### *T. cruzi* infection modulates hiPSC-CM ST8Sia2 expression and compromises host polysialylation

In order to investigate the impact of host polysialylation upon *T. cruzi* infection, we differentiated human pluripotent stem cells (hiPSC) into cardiomyocytes (hiPSC-CM), and the efficiency of differentiation on hiPSC-CM was determined by the expression of troponin T and α-actinin, using immunofluorescence microscopy (**S2A Fig**), and by the percentage of positive cells stained with anti-troponin T (TNNT2+) antibodies analyzed by flow cytometry (**S2B Fig**). Then, we evaluated the levels of polysialylation in hiPSC-CM infected with *T. cruzi* (**Fig 1**). *T. cruzi* parasites were efficiently detected 48 h.p.i. (**Fig 1A and 1B**), as described previously [50]. Corroborating our initial data, we observed a 70% downregulation of *ST8Sia2* mRNA in *T. cruzi*-infected hiPSC-CM using qRT-PCR (**Fig 1C**). Furthermore, ST8Sia2 protein levels are also reduced in infected hiPSC-CM (**Fig 1D and 1H**).

Since polysialylation is a post-translational modification restricted to a set of glycoproteins [16], including the Sodium channel protein type 5 subunit alpha (SCN5A) of the primary cardiac isoform of the voltage-gated sodium channel (Na$_V$1.5), and neural cell adhesion molecule 1 (NCAM1) [51], both involved in modulating cardiac functions [29,52], both molecules were focused herein. Following the reduction in ST8Sia2 levels observed in **Fig 1**, reduced polySia levels were also detected in *T. cruzi*-infected hiPSC-CM (**Fig 1E and 1H**). No significant modulation in NCAM1 levels was observed between hiPSC-CM infected and non-infected with *T. cruzi* (**Fig 1F and 1H**), as expected since the cardiomyocyte model used in this study did not express PSA-NCAM, the polysialylated form of NCAM1 (**S2C Fig**). Accordingly, no signal was detected using a specific antibody for PSA-NCAM in non-infected (**S2C Fig; lane 1**) and infected (**S2C Fig; lane 2**) hiPSC-CM. As controls, the efficiency of immunostaining was demonstrated in TE671 cells (**S2C Fig; lane 3**), a human rhabdomyosarcoma cell previously described as positive for NCAM and PSA-NCAM [53] specificity of the antibody towards host polysialylation was demonstrated using *T. cruzi* trypomastigotes (**S2C Fig; lane 4**). In addition, we detected a reduction in SCN5A levels in *T. cruzi*-infected hiPSC-CM (**Fig 1G and 1H**). These data suggested that infection of hiPSC-CM with *T. cruzi* leads to the downregulation of ST8Sia2, affecting polySia transfer to specific substrates such as SCN5A, thereby reducing their abundance.

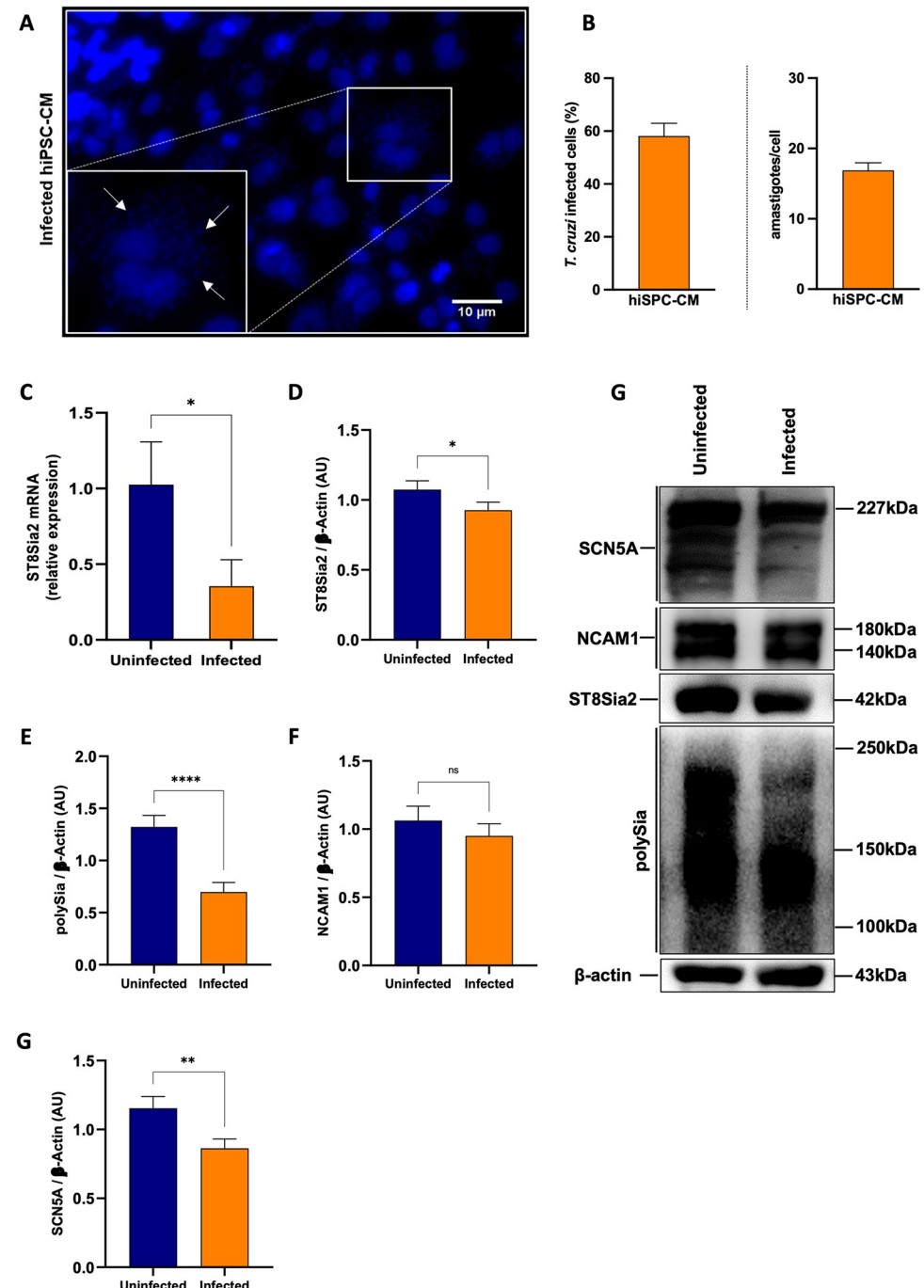

**Fig 1. *T. cruzi* infection modulates the abundance of ST8Sia2, SCN5A, and polysialylation on hiPSC-CM. (A-G)** hiPSC-CM cells were seeded in 24-well microplates (2 x 10^5 cells/well) and infected with *T. cruzi* trypomastigotes (Y strain) at ratio of 1:5 (hiPSC-CM:trypomastigotes), follow by 48 h incubation at 37˚C. hiPSC-CM cells incubated with medium alone (Uninfected) were used as negative control; **(A)** Representative immunofluorescence images of *T. cruzi*-infected hiPSC-CM. Cell nuclei were stained with DAPI (scale bars = 10 μm). White arrows are representative of *T. cruzi* amastigotes inside hiPSC-CM cells; **(B)** Representative graphs of the percentage of *T. cruzi*-infected hiPSC-CM (left), and quantification of the number of intracellular parasites (amastigotes) per infected cell (right); **(C)** Relative expression of mRNA *ST8Sia2* measured by qRT-PCR in *T. cruzi*-infected hiPSC-CM. The Ct values of the target transcripts were normalized to the relative expression of *GAPDH* as endogenous control, and the relative expression of *ST8Sia2* transcripts was quantified by the $2^{-\Delta\Delta}$ Ct method. Each bar represents the mean of three independent experiments performed in triplicate; **(D-H)** Representative Western blotting quantification of ST8Sia2 **(D)**, polySia **(E)**, NCAM1 **(F)**, and SCN5A **(G)** in *T. cruzi*-infected hiPSC-CM cells. The protein levels were analyzed by Western

blotting of RIPA cell lysates (15 μg protein) performed under reducing conditions. Results are presented as arbitrary densitometry units (AU). After normalization to the corresponding β-actin content (endogenous control), data were plotted as the ratio between the values obtained in infected and uninfected cells in their respective endogenous controls. Each bar represents the mean of three independent experiments performed in triplicate; **(H)** Representative Western blotting images of levels of ST8Sia2 **(D)**, polySia **(E)**, NCAM1 **(F)**, and SCN5A **(G)** in *T. cruzi*-infected hiPSC-CM cells. Results are expressed as mean ± SEM. Significant differences compared to the uninfected cells are shown by (*) $p < 0.05$, (**) $p < 0.001$, and (***) $p < 0.0001$; ns = not significant.

## *T. cruzi* infection reduces the expression of host polysialylated target molecules

In order to broaden the approach to the effect of *T. cruzi* infection on the host polysialylation machinery, it became interesting to investigate the impact on polysialylation in SH-SY5Y cells, a human cell line of neuroblastoma with high expression of ST8Sia2. The infection efficiency of SH-SY5Y cells with *T. cruzi* was evaluated after 48 h.p.i by quantifying internalized parasites and determining the percentage of infected cells (**Fig 2A and 2B**). Next, we investigated the impact of *T. cruzi* infection on the polysialylation of SH-SY5Y cells (**Fig 2C, 2D and 2H**). Similar to the findings observed on *T. cruzi*-infected hiPSC-CM, we detected that ST8Sia2 (**Fig 2C and 2H**), polySia (**Fig 2D and 2H**), and SCN5A (**Fig 2E and 2H**) levels were significantly reduced in *T. cruzi*-infected SH-SY5Y cells. We also observed reduced levels of NCAM1 (**Fig 2F and 2H**) in SH-SY5Y cells infected with *T. cruzi*. Furthermore, there was a significant reduction in the abundance of PSA-NCAM (**Fig 2G and 2H**). Thus, these findings reinforce our hypothesis that the host cell polysialylation is modulated during the course of *T. cruzi* infection.

Furthermore, we evaluated the abundance of ST8Sia2 and polySia in *T. cruzi*-infected SH-SY5Y cells using immunofluorescence microscopy approach. Corroborating the data presented in **Fig 2**, we observed a significant reduction in the abundance of ST8Sia2 (**Fig 3A and 3C**) and polySia (**Fig 3B and 3D**) in *T. cruzi*-infected SH-SY5Y cells, when compared to non-infected cells. Next, we quantified the polySia content in the supernatant of SH-SY5Y cells infected with *T. cruzi* using a chromatographic method with fluorescence-based detection. For this assay, SH-SY5Y cells infected and non-infected with *T. cruzi* were incubated with endo-neuraminidase (EndoN), an enzyme that specifically cleaves α2,8-linked sialic acid polymers [16]. We observed that the polySia levels from the supernatant of *T. cruzi*-infected SH-SY5Y cells were significantly reduced compared to non-infected SH-SY5Y cells (**Fig 3E**), reinforcing our findings that host polysialylation is differentially modulated upon *T. cruzi* infection. In addition, we treated SH-SY5Y cells with EndoN to remove polySia and evaluated whether it would have an impact on *T. cruzi* infection, adopting the workflow shown in **Fig 4A**. Our results showed that removal of polySia in SH-SY5Y cells increased the number of intracellular parasites 48 h.p.i (**Fig 4B and 4C**), reinforcing that host polysialylation has a vital role during *T. cruzi* infection.

## Chemical and genetic Inhibition of ST8Sia2 increases the number of parasites internalized in SH-SY5Y cells

The impact of *T. cruzi* infection on the modulation of host polysialylation was demonstrated here by different experimental approaches in **Figs 1–4**. Thereafter, we evaluated whether chemical and genetic inhibition of ST8Sia2 would have an impact on host cell infection by *T. cruzi*. For this, we used cytidine 5′-monophosphate (CMP), a molecule previously reported as a pharmacological inhibitor of ST8Sia2 [54], and the siRNA-directed approach. Initially, we demonstrated that chemical or genetic inhibition of ST8Sia2 in SH-SY5Y cells with CMP or

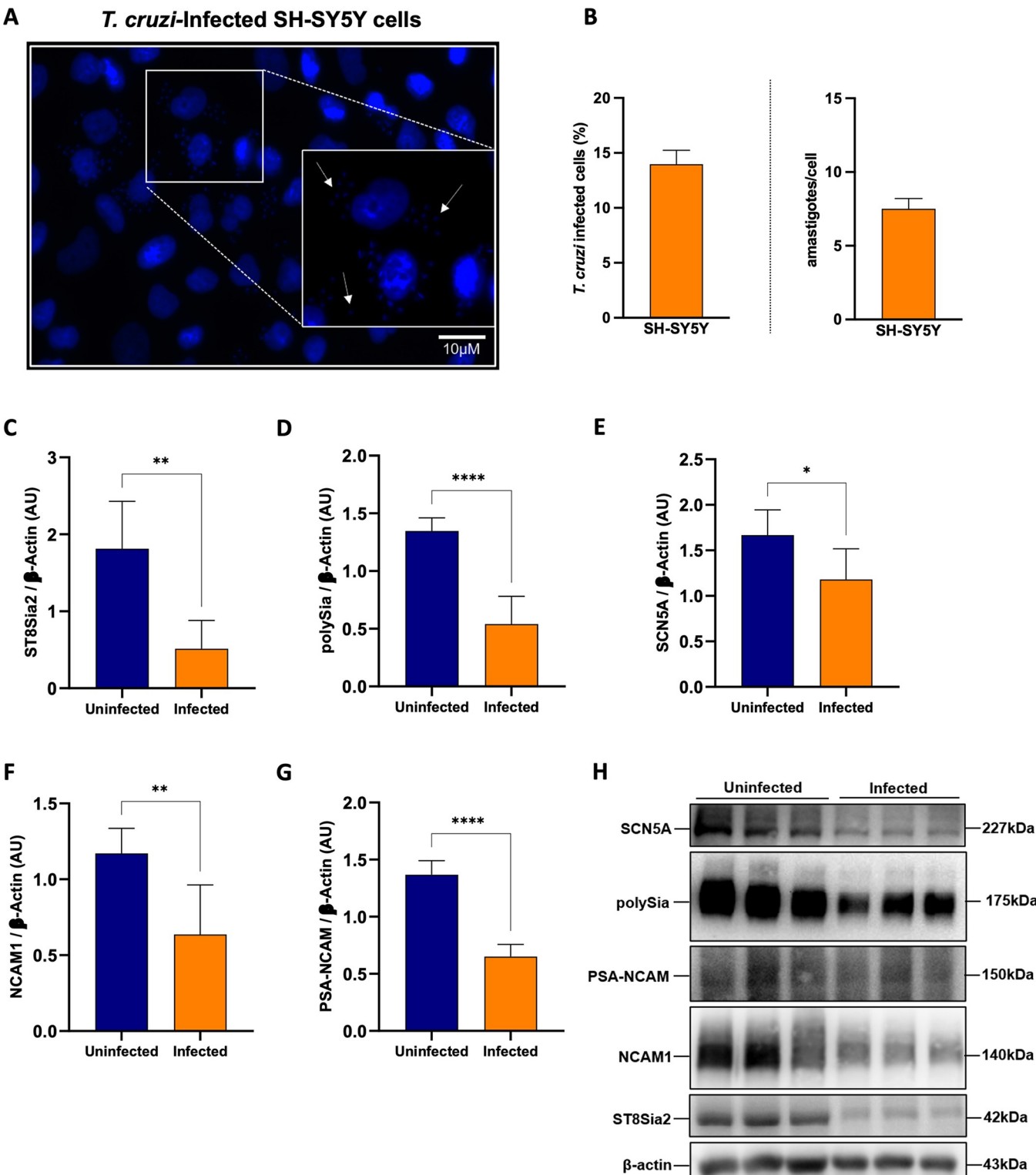

**Fig 2. *T. cruzi* infection modulates the polysialylation on SH-SY5Y cells and affects the abundance of polysialylated molecules. (A)** Representative immunofluorescence images of *T. cruzi*-infected SH-SY5Y cells. SH-SY5Y cells were seeded in 24-well microplates (5 x 10$^5$ cells/well) and infected with *T. cruzi* trypomastigotes (Y strain) at ratio of 1:5 (SH-SY5Y:trypomastigotes) at 37˚C. 48 h.p.i SH-SY5Y cells incubated with medium alone (Uninfected) was used as negative control. Cell nuclei were stained with DAPI (scale bars = 10 μm). White arrows are representative of *T. cruzi* amastigotes inside in SH-SY5Y cells; **(B)** Representative graphs of the percentage of *T. cruzi*-infected hiPSC-CM (left), and quantification of the number of intracellular parasites (amastigotes) per

infected cell (right); **(C-H)** Western blotting quantification of ST8Sia2 levels **(C),** polysialic acid (polySia) **(D)**, SCN5A **(E)**, NCAM1 **(F)** and PSA-NCAM **(G)** in *T. cruzi*-infected SH-SY5Y cells. SH-SY5Y cells were seeded in 6-well microplates (1 x 10⁶ cells/well) and infected with *T. cruzi* as in **A**. 48 h.p.i, the proteins levels were analyzed by Western blotting of RIPA cell lysates (15 μg protein) performed under reducing conditions. Results are presented as arbitrary densitometry units (AU). After normalization to the corresponding β-actin content (endogenous control), data were plotted as the ratio between the values obtained in infected and uninfected cells in their respective endogenous controls. Each bar represents the mean of three independent experiments performed in triplicate; **(H)** Representative Western blot images of ST8Sia2 **(C)**, polySia **(D)**, SCN5A **(E)**, NCAM1 **(F)**, and PSA-NCAM **(G)** in *T. cruzi*-infected SH-SY5Y cells. Results are expressed as mean ± SEM. Significant differences compared to the uninfected cells are shown by (*) $p < 0.05$, (**) $p < 0.001$, and (****) $p < 0.0001$.

siRNA *ST8Sia2*, respectively, does not affect cell viability (**S3A and S3B Fig**). Then, SH-SY5Y cells were treated with CMP and infected with *T. cruzi* according to the workflow shown in **Fig 5A**. The percentage of infected cells, and the number of internalized parasites was determined after labeling them at 48 h post-infection (**Fig 5B and 5C**). We demonstrated that chemical inhibition of ST8Sia2 with CMP in SH-SY5Y cells increased the number of intracellular parasites (**Fig 5C and 5F**). We also demonstrated that the percentage of infected cells and the number of intracellular parasites in untreated cells (culture medium only) or GMP-treated cells did not differ significantly (**Fig 5C and 5F**), demonstrating that the inhibition of ST8Sia2 favors infection by *T. cruzi*. Along the same purpose, we used a siRNA-directed approach to silence *ST8Sia2* according to the workflow presented in **Fig 5A**. Transfection efficiency was evaluated by the relative expression of the *ST8Sia2* transcript by qRT-PCR (**S3C Fig**) according to the workflow shown in **S3D Fig**. Additionally, we validated the *ST8Sia2* silencing strategy in SH-SY5Y cells by visualizing and quantifying polySia levels using immunofluorescence microscopy (**S3E Fig**) and UHPLC approach (**S3F Fig**), respectively. Our results demonstrated that genetic silencing of *ST8Sia2* in SH-SY5Y cells also increased the percentage of infected cells (**Fig 5D**), and the load of internalized parasites (**Fig 5E and 5G**). Taken together, these findings reinforce our hypothesis that ST8Sia2 inhibition favors *T. cruzi* infection.

## Silencing of *ST8Sia2* in *T. cruzi*-infected SH-SY5Y cells intensifies oxidative stress

Herein, we have utilized different experimental techniques and strategies to demonstrate the impact of *T. cruzi* infection on the modulation of host cell polysialylation. However, insights into a possible mechanism where the effect of polySia could influence the cell survival of *T. cruzi* remain unknown. It has been previously demonstrated that in *T. cruzi* infection, the parasite requires signaling provided by reactive oxygen species (ROS) from macrophages to establish a successful infection [55]. In HeLa cells, oxidative stress and DNA damage are activated during *T. cruzi* infection. Cho *et al*. [56] reported that sialic acid deficiency is associated with oxidative stress. In this study, it was observed that lower levels of sialic acid increase oxidative stress.

Considering these evidence collectively, we hypothesize that the impact on host polysialylation, and consequently, the increase in parasite survival could be related to the imbalance of oxidative stress. To confirm our hypothesis that there is an potential interplay between the disruption in polysialylation machinery and a generation of a status of oxidative stress, we used the strategy of *ST8Sia2* siRNA-directed silencing (**S3D Fig**), and monitored oxidative stress in the presence or absence of Antimycin A and/or N-acetyl cysteine, both molecules previously reported as inducer [57] or inhibitor [58] of oxidative stress, respectively.

When measuring cytosolic and mitochondrial ROS levels in SH-SY5Y si*ST8Sia2* cells, we observed a significant reduction in cytosolic (**Fig 6A**) and mitochondrial (**Fig 6D**) ROS production, compared to the baseline levels detected in SH-SY5Y cells from the control group (NTC). We also demonstrated that in response to treatment with antimycin A, SH-SY5Y

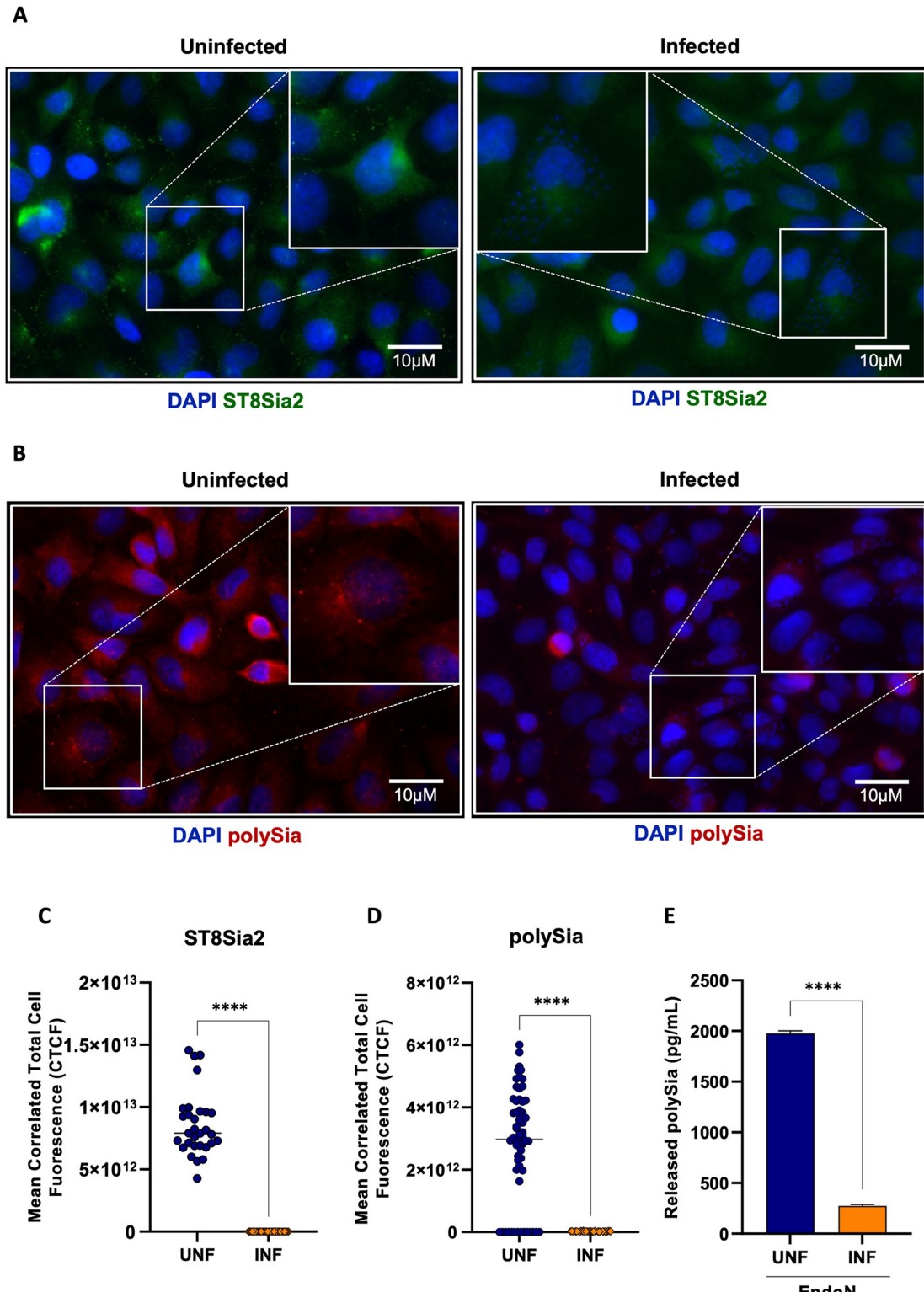

**Fig 3. *T. cruzi* infection reduces ST8Sia2 and PolySia levels in SH-SY5Y cells. (A,B)** Representative images of ST8Sia2 (**A**) and PolySia (**B**) levels in *T. cruzi*-infected SH-SY5Y cells. SH-SY5Y cells were seeded in 24-well microplates (5 x 10⁵ cells/well) and infected with *T. cruzi* trypomastigotes (Y strain) at ratio of 1:5 (SH-SY5Y:trypomastigotes) at 37˚C. After 48 h.p.i, *T. cruzi*-infected SH-SY5Y cells were stained with specific anti-ST8Sia2 (**A**) and anti-polySia (**B**) antibodies, and labeling for the targets was visualized by immunofluorescence microscopy. SH-SY5Y cells incubated with medium alone (Medium) was used as negative

controls. Cell nuclei were stained with DAPI (scale bars = 10 μm); **(C,D)** Quantification of ST8Sia2 **(C)** and polySia **(D)** fluorescence intensity in *T. cruzi*-infected SH-SY5Y cells using the calculation for corrected total cell fluorescence (CTCF) as explained in Material and Methods section. Each dot represents the CTCF read out from one cell. A total of 50 SH-SY5Y cells per condition (infected and uninfected) were quantified. The values are expressed as Mean CTCF ± standard error of the mean (SEM); **(E)** polySia levels measured in *T. cruzi*-infected and non-infected SH-SY5Y cells supernatants. SH-SY5Y cells were seeded in T75-flasks (5 x $10^5$ cells/mL) and infected with *T. cruzi* trypomastigotes (Y strain) at ratio of 1:5 (SH-SY5Y:trypomastigotes) at 37°C. After 48 h.p.i, *T. cruzi*-infected SH-SY5Y cells were treated with EndoN [0.5 μg/ml] for 1h at 37°C, and polySia levels were measured in supernatants using UHPLC. The same procedure was applied to non-infected SH-SY5Y cells. The results are expressed in pg/mL. Significant differences compared to the uninfected are shown by (****) $p < 0.0001$.

si*ST8Sia2* cells exhibited higher levels of cytosolic (**Fig 6B**) and mitochondrial (**Fig 6E**) ROS. On the other hand, no significant change was observed in cytosolic (**Fig 6C**) and mitochondrial (**Fig 6F**) ROS levels in SH-SY5Y si*ST8Sia2* cells treated with N-acetyl cysteine, compared to the levels detected in SH-SY5Y NTC cells. These results provide for the first time that modulation of host polysialylation can influence oxidative stress.

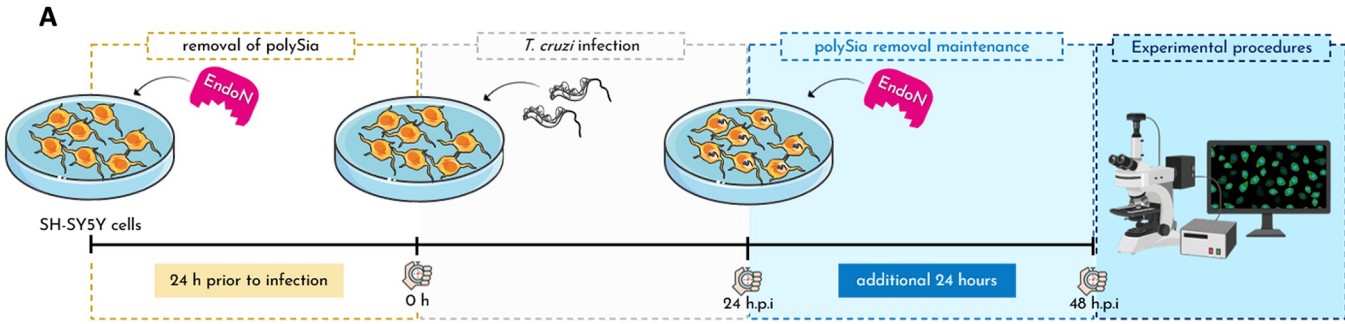

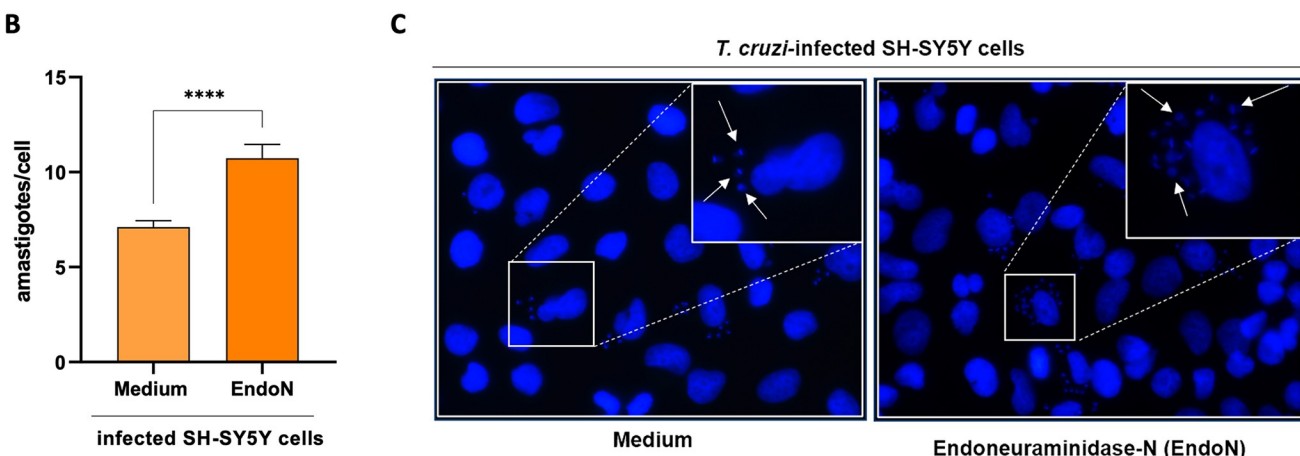

**Fig 4. Removal of polySia in SH-SY5Y cells favors *T. cruzi* infection by increasing the number of internalized parasites. (A)** Experimental workflow adopted to investigate the effect of enzymatic removal of polySia in *T. cruzi*-infected SH-SY5Y cells by EndoNeuraminidase (EndoN) [0.5 μg/ml]. SH-SY5Y cells were seeded in 24-well microplates (5 x $10^5$ or 5 x $10^4$ cells/well) and treated with EndoN [0.5 μg/ml] for 24h before infection, followed by addition of *T. cruzi* trypomastigotes (Y strain) at ratio of 1:5 (SH-SY5Y:trypomastigotes). After 24 h.p.i, *T. cruzi*-infected SH-SY5Y cells were submitted to another treatment with EndoN [0.5 μg/ml] for an additional 24h. Medium alone (Medium) were used as negative control; **(B)** Quantification of the number of intracellular parasites (amastigotes) per infected cell. Cell nuclei were stained with DAPI and following the experimental approach shown in **B**, amastigotes were counted in a total of 100 infected SH-SY5Y cells. Results are expressed as mean ± standard error of the mean (SEM) performed in triplicates; **(C)** Representative immunofluorescence images of *T. cruzi*-infected SH-SY5Y cells following the strategy adopted in the experimental workflow presented in **B**. Cell nucleus were stained with DAPI (scale bars = 10 μm). White arrows are representative of *T. cruzi* amastigotes inside in SH-SY5Y cells. Significant differences compared to the *T. cruzi*-infected SH-SY5Y treated with medium alone (Medium) are shown by (****) $p < 0.0001$; *ns*: not significant. Parts of the figure were drawn by using pictures from Servier Medical Art. Servier Medical Art by Servier is licensed under a Creative Commons Attribution 3.0 Unported License (https://creativecommons.org/licenses/by/3.0/), and BioRender.com under academic license.

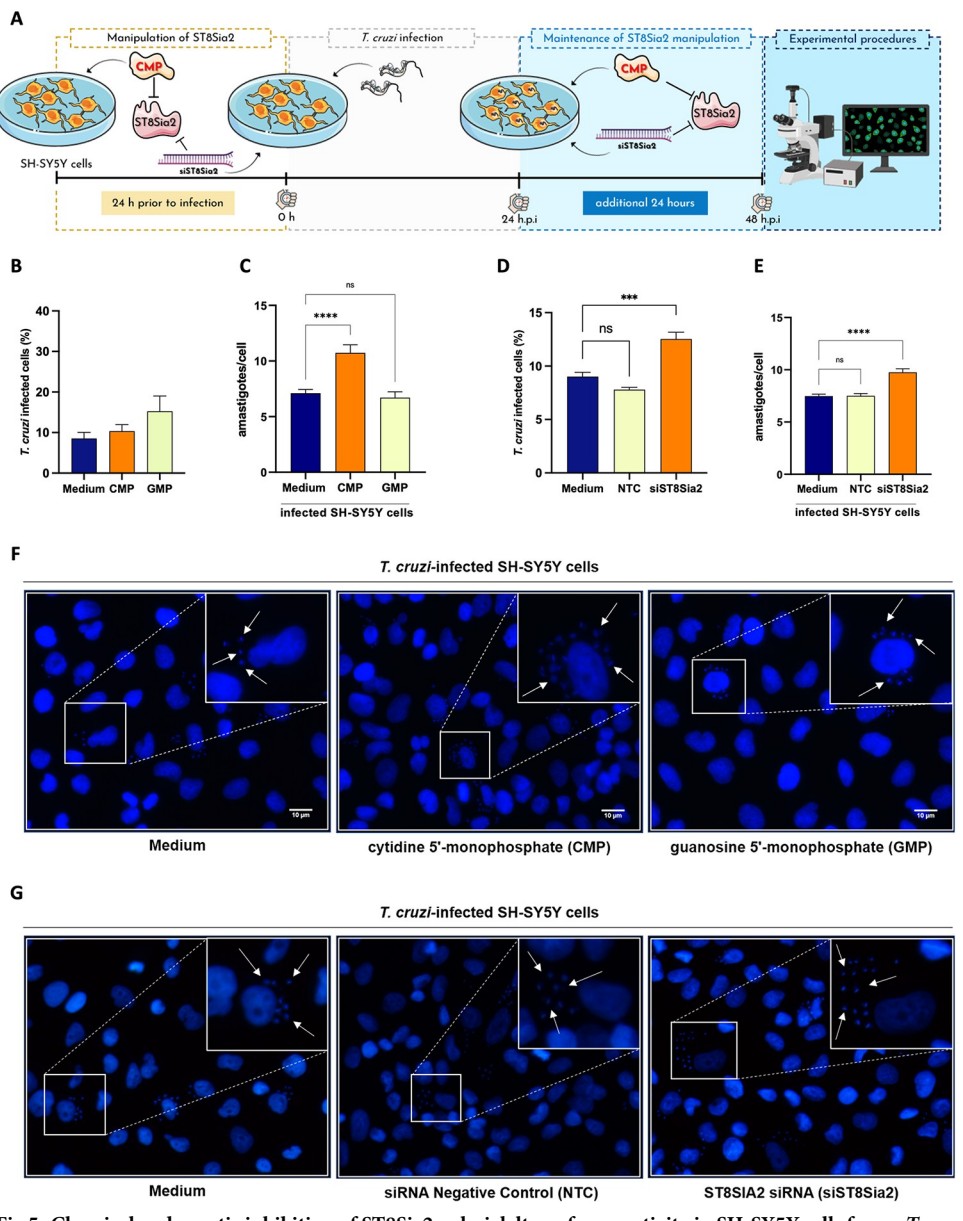

**Fig 5. Chemical and genetic inhibition of ST8Sia2 polysialyltransferase activity in SH-SY5Y cells favors *T. cruzi* proliferation.** (**A**) Experimental workflow adopted to investigate the effect of chemical and genetic inhibition of ST8Sia2 in *T. cruzi*-infected SH-SY5Y cells using 0.5 mM of CMP. SH-SY5Y cells were seeded in 24-well microplates ($5 \times 10^5$ or $5 \times 10^4$ cells/well), and previously treated with CMP [0.5 mM] or siRNA *ST8Sia2* [100 nM] for 24h before infection, followed by infection with *T. cruzi* trypomastigotes (Y strain) at ratio of 1:5 (SH-SY5Y:trypomastigotes). After 24 h.p.i, *T. cruzi*-infected SH-SY5Y cells were treated with CMP [0.5 mM] or siRNA *ST8Sia2* [100 nM] for an additional 24h. Guanosine 5'-monophosphate (GMP), siRNA negative control (NTC), and/or medium alone (Medium) were used as negative controls for chemical or genetic inhibition of *ST8Sia2*. In all controls used in experiments with siRNA was added Lipofectamine RNAiMAX; (**B**) Representative graphs of the percentage of *T. cruzi*-infected SH-SY5Y treated with CMP; (**C**) Quantification of the number of intracellular parasites (amastigotes) per infected cell treated with CMP. Cell nuclei were stained with DAPI and following the experimental approach shown in **A**, amastigotes were counted in a total of 100 infected SH-SY5Y cells. (**D**) Representative graphs of the percentage of *T. cruzi*-infected SH-SY5Y treated with siRNA *ST8Sia2* (siST8Sia2); (**E**) Quantification of the number of intracellular parasites (amastigotes) per infected cell. Cell nuclei were stained with DAPI and following the experimental approach shown in **A**, amastigotes were counted in a total of 100 infected SH-SY5Y cells. Results are expressed as mean ± standard error of the mean (SEM) performed in triplicates; (**F,G**) Representative immunofluorescence images of *T. cruzi*-infected SH-SY5Y cells treated with CMP (**F**), and *T. cruzi*-infected SH-SY5Y treated with siRNA *ST8Sia2* (**G**), following the strategy adopted in the experimental workflow presented in **A**. Cell nuclei were stained with DAPI (scale bars = 10 μm). White arrows are representative of *T. cruzi* amastigotes inside in SH-SY5Y cells. Significant

differences compared to the *T. cruzi*-infected SH-SY5Y treated with medium alone (Medium) are shown by (***) *p < 0.0005*, (****) *p < 0.0001*; *ns*: not significant. Parts of the figure were drawn by using pictures from Servier Medical Art. Servier Medical Art by Servier is licensed under a Creative Commons Attribution 3.0 Unported License (https://creativecommons.org/licenses/by/3.0/), and BioRender.com under academic license.

Under conditions of oxidative stress, alterations in intracellular calcium ion ($Ca^{2+}$) levels are frequently observed [59]. Elevated intracellular $Ca^{2+}$ concentration can modulate nitric oxide (NO) production, culminating in the generation of oxidants, thereby amplifying $Ca^{2+}$ levels and subsequently exacerbating oxidative stress [60]. It has been documented that in conditions of oxidative stress, the concurrent decrease in mitochondrial membrane potential ($\Delta\Psi$m) coupled with heightened $Ca^{2+}$ production signify potential mitochondrial dysfunction and oxidative stress [61]. Subsequently, mitochondrial membrane potential and mitochondrial $Ca^{2+}$ levels were assessed in SH-SY5Y si*ST8Sia2* cells. We observed that silencing of *ST8Sia2* led to dysregulation in mitochondrial membrane potential (**Fig 6G**), concomitant with a marked elevation in $Ca^{2+}$ production (**Fig 6H**), as compared to baseline levels detected in SH-SY5Y cells from the control group (NTC). Taken together, these findings strongly suggest that the reduced levels of ST8Sia2 alter the baseline levels of factors implicated in oxidative stress, notably ROS and $Ca^{2+}$ intracellular, potentially heightening oxidative stress reactivity upon exposure to secondary stimuli such as Antimycin A.

Additionally, a bottom-up large-scale mass spectrometry-based proteomics approach was employed to identify and quantify proteins differentially regulated in *T. cruzi*-infected *ST8Sia2*-silenced SH-SY5Y cells (si*ST8Sia2 Ty*). A total of 851 proteins were identified in this analysis, of which 762 were quantified across all biological replicates (**Fig 7A, left flowchart, S1 and S2 Tables**), and 24 proteins were quantified exclusively (**Fig 7A, right flowchart**) in *T. cruzi*-infected si*ST8Sia2* SH-SY5Y cells and *T. cruzi*-infected SH-SY5Y cells incubated with culture medium alone (**Fig 7A**). A clear separation into two large groups was observed based on the Principal Component Analysis (PCA) (**Fig 7B**).

A total of 68 proteins were regulated between *T. cruzi*-infected si*ST8Sia2* SH-SY5Y cells and *T. cruzi*-infected SH-SY5Y cells incubated with culture medium alone (S3 Table). In the Chord plot shown in **Fig 7C**, we mapped in the total dataset the expression of proteins involved in pathways and biological processes related to the stress response. We identified 62 proteins whose associated with pathways and biological processes related to oxidative stress response, stress granule assembly, and abiotic and biotic stress (**Fig 7C**). These proteins showed a modulation in *T. cruzi*-infected *ST8Sia2*-silenced (si*ST8Sia2 Ty*) compared to non-silenced (Medium *Ty*) SH-SY5Y cells (**Fig 7C**). After applying statistical test, among the differentially regulated proteins (**Fig 7D**), we identified a set of proteins involved in oxidative stress such as the upregulation of proteolipid protein 2 (PLP2, Q04941) (**Fig 7D and 7E**) and protein S100-A6 (P06703) (**Fig 7D and 7F**), and the downregulation of serine/threonine-protein kinase PAK 2 (Q13177) (**Fig 7D and 7G**) and DNA mismatch repair protein MSH2 (P43246) (**Fig 7D and 7H**) in *T. cruzi*-infected si*ST8Sia2* SH-SY5Y cells. Collectively, these results reinforce our hypothesis that the infection rate of *T. cruzi* in host cells is enhanced by the downregulation of *ST8Sia2* and the reduction in polysialylation, which is linked to increased oxidative stress that creates a microenvironment permissive to infection.

## Discussion

Different studies have extensively explored the manipulation of host cell glycosylation in infectious processes since many pathogens use this post-translational modification to ensure a successful infection [62,63]. In the case of *T. cruzi*, the role of Sia in the infection has been

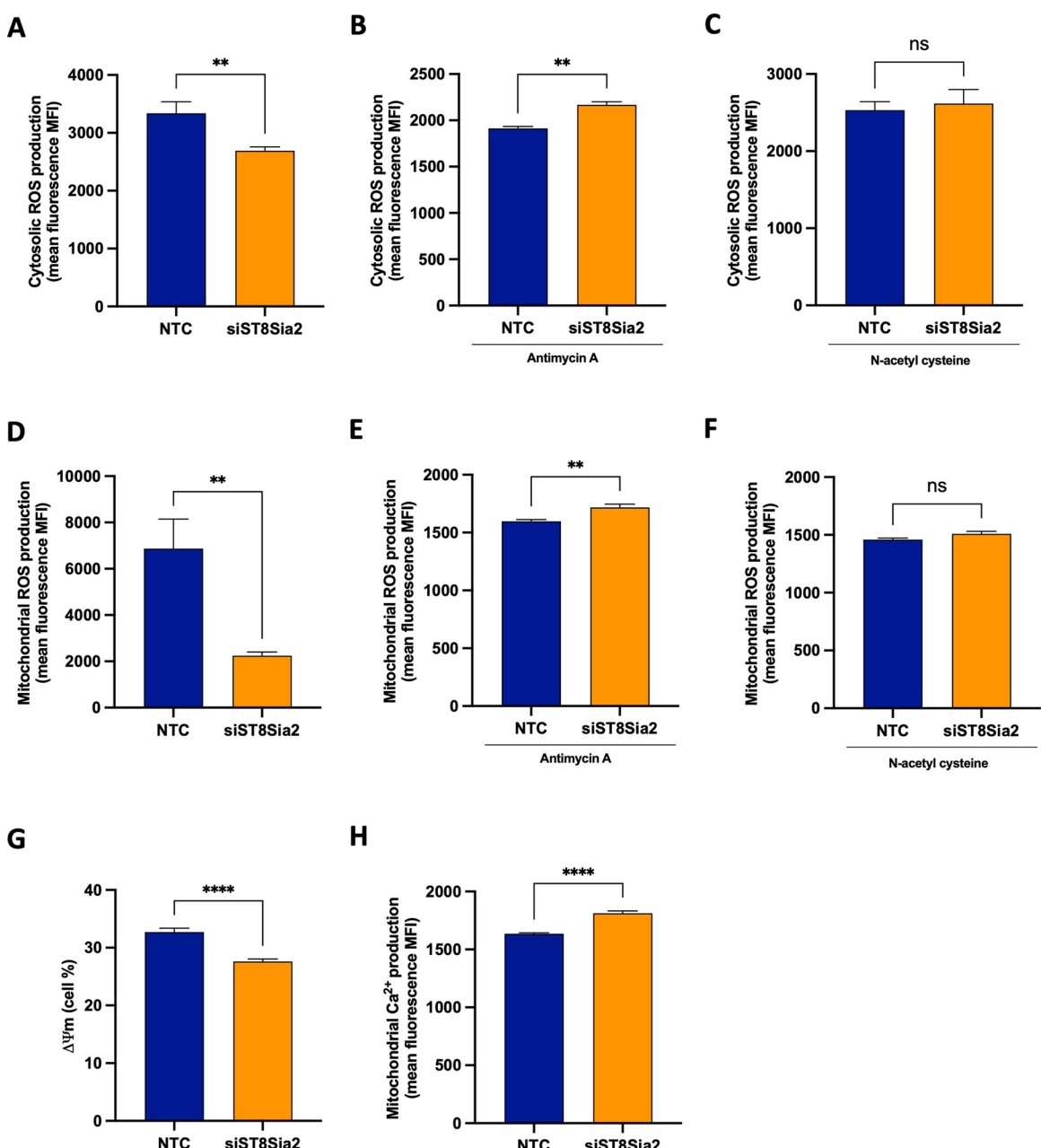

**Fig 6. Silencing of *ST8Sia2* in SH-SY5Y cells modulates oxidative stress.** SH-SY5Y cells were seeded in 6-well microplates ($1 \times 10^6$ cells/well) and treated with siRNA *ST8Sia2* [100 nM] for 24 hours. After this time, SH-SY5Y si*ST8Sia2* cells were treated or not with Antimycin A [50 μM] or N-acetyl cysteine [50 μM] for 4 hours. Negative control siRNA (NTC) was used as the negative control for silencing of *ST8Sia2*. In all controls used in experiments with siRNA was added Lipofectamine RNAiMAX. **(A-C)** SH-SY5Y si*ST8Sia2* cells previously treated or not with Antimycin A and/or N-acetyl cysteine were incubated with Dihydroethidium [3 μM] at 37°C for 40 minutes to determine cytosolic reactive oxygen species (ROS) levels; **(D-F)** SH-SY5Y si*ST8Sia2* cells previously treated or not with Antimycin A and/or N-acetyl cysteine were incubated with MitoSOX Red [5 μM] at 37°C for 10 minutes to determine mitochondrial reactive oxygen species (ROS) levels; **(G)** SH-SY5Y si*ST8Sia2* cells were incubated with MitoTracker Red CMXRos Dye [100 nM] at 37°C for 30 minutes to determine mitochondrial membrane potential (ΔΨm); **(H)** SH-SY5Y si*ST8Sia2* cells were incubated with Rhod-2 [5 μM] at 37°C for 60 minutes to determine mitochondrial calcium ion ($Ca^{2+}$) levels. Results are expressed as mean ± SEM. Significant differences compared to SH-SY5Y cells treated with negative control siRNA (NTC) are indicated by (**) $p < 0.005$, (****) $p < 0.0001$; ns: not significant.

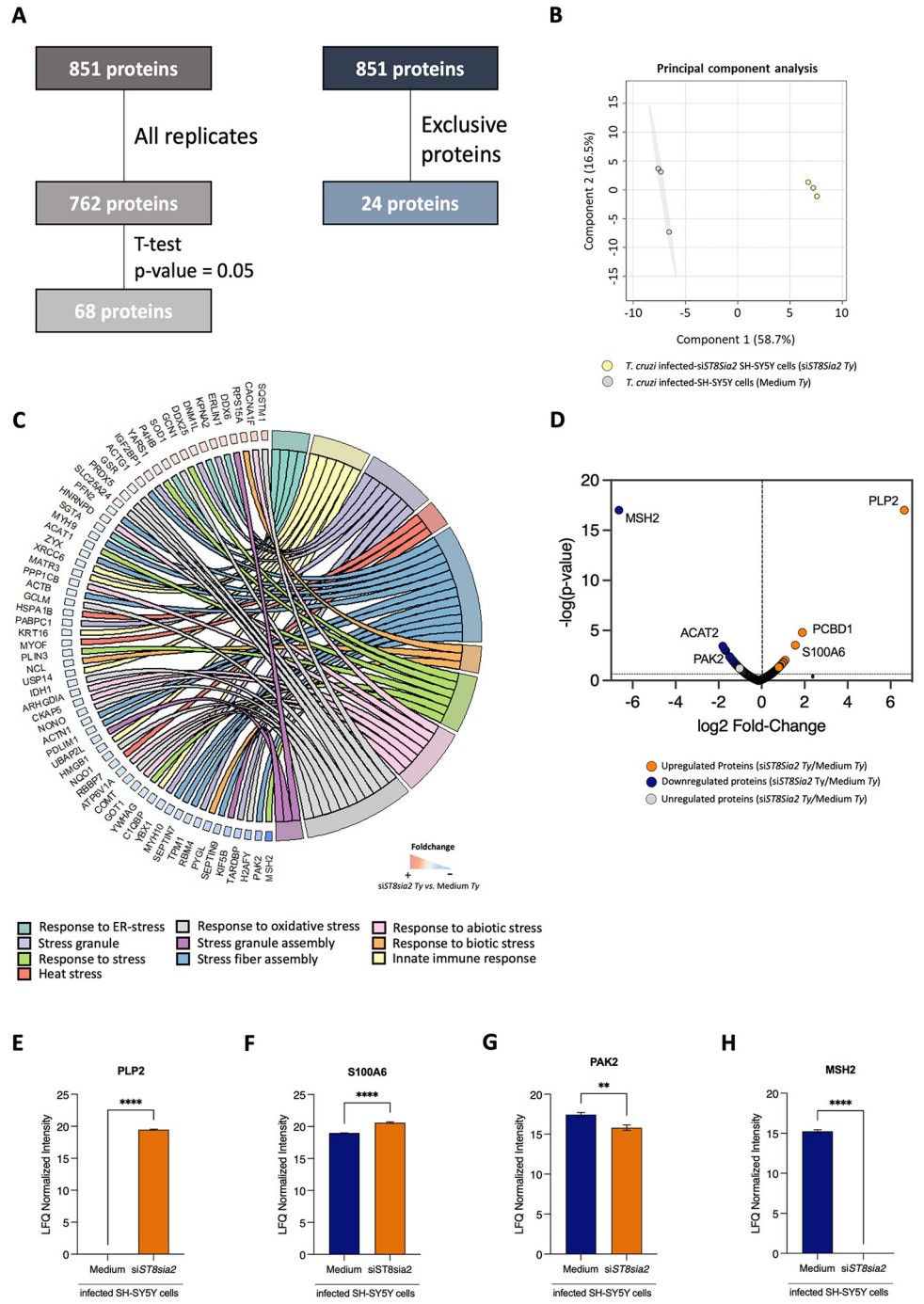

**Fig 7. Proteomic analysis of SH-SY5Y cells with *ST8sia2* gene silencing (si*ST8sia2*) and Lipofectamine RNAiMAX-treated control (Medium), both infected with *T. cruzi* (*Ty*).** SH-SY5Y cells were seeded in 6-well microplates (1 x 10^6 cells/well), and previously treated with siRNA *ST8Sia2* [100 nM] for 24h before infection, followed by infection with *T. cruzi* trypomastigotes (Y strain) at ratio of 1:5 (SH-SY5Y:trypomastigotes). After 24 h.p.i, *T. cruzi*-infected SH-SY5Y cells were treated with siRNA *ST8Sia2* [100 nM] for an additional 24h. Medium alone (Medium) was used as negative control for genetic inhibition of *ST8Sia2*. In all controls used in experiments with siRNA was added Lipofectamine RNAiMAX. **(A)** Proteins identified in all biological replicates or exclusively in one of the evaluated groups: cells silenced for *ST8sia2* (si*ST8sia2*, n = 3) and medium-treated with Lipofectamine RNAiMAX (Medium, n = 3); **(B)** Principal component analysis (PCA) of SH-SY5Y cells silenced for *ST8sia2* (si*ST8sia2*, green) and Medium-treated control (Medium, grey), both infected with *T. cruzi*; **(C)** Chord plot showing the proteins involved in biological pathways and processes related to the different stress responses; **(D)** Volcano plot illustrating the differentially regulated proteins in the *T. cruzi*-infected si*ST8si2* SH-SY5Y *vs*. Medium comparison, with upregulated

proteins shown in orange, downregulated proteins in blue, and unregulated proteins in grey ($p$-value $< 0.05$); **(E)** Boxplot of differentially regulated stress response proteins in *T. cruzi*-infected SH-SY5Y Cells with *ST8Sia2* Silencing. Significant differences compared to *T. cruzi*-infected si*ST8Sia2* SH-SY5Y cells with *T. cruzi*-infected SH-SY5Y cells treated with medium (negative control for genetic silence of *ST8Sia2* are indicated by p $<0.0001$ (\*\*\*\*); p $<0.001$ (\*\*\*); p $<0.01$ (\*\*); p $< 0.05$ (\*).

extensively explored, with a focus on the transfer of this sugar from host α-2,3-sialylated glycans to the surface of the parasite catalyzed by trans-sialidase [37,40]. Also, knockout mice for *CD43* [48], a sialoglycoprotein previously reported as a natural receptor for *T. cruzi* trans-sialidase [47], showed reduced cardiac parasite complications [48], reinforcing that manipulation of host glycosylation could directly or indirectly interfere in the *T. cruzi* infection. However, the impact of *T. cruzi* infection on host glycosylation machinery has not been explored to date.

Herein, we described the modulation of host cell N-glycosylation machinery in experimental *T. cruzi* infection, and, providing for the first time, evidence of downregulation of *ST8Sia2*, a host polysialyltransferase. Using an initial *in-silico* approach, different genes encoding enzymes involved in the protein glycosylation pathways, was suggested to be up (e.g. Neuraminidase 1—*Neu1*, and α-1,3-mannosyltransferase—*ALG3*, or down (ST8 alpha-N-acetyl-neuraminide alpha-2,8-sialyltransferase 2 –*ST8Sia2*, and α-Glucosidase 2—*GANAB*) regulated in the course of infection in human cardiomyocytes (**S1 Fig**). Among these (glyco)genes, *ST8Sia2* showed a sustained downregulation over different time points (**S1 Fig**), which should lead at the end to a decrease in the polySia content of glycoproteins, as described in the literature [64].

Consistent with our initial observations, we found experimentally a significant reduction in *ST8Sia2* mRNA levels (**Fig 1C**) and polysialic content (**Fig 1E and 1H**) in human induced-pluripotent stem cells derived cardiomyocyte (hiPSC-CM) after 48 h of *T. cruzi* infection, although a less expressive decrease at the protein level of ST8Sia2 was observed at this time point (**Fig 1D and 1H**). Similar results, regarding decrease of ST8Sia2 expression (**Fig 2C and 2H**) and polySia (**Fig 2D and 2H**) were obtained with SH-SY5Y, a human neuroblastoma cell line, after 48 h of *T. cruzi* infection. A direct correlation between the expression of ST8Sia2 and the reduction in polySia levels has already been well demonstrated in previous studies that investigated the relevance of polysialylation in the nervous system [65,66].

Although polySia is added to N- and O-glycosylation structures, its presence is confined to a restricted set of glycoproteins, emphasizing NCAM1, which is the main and best-studied carrier of polySia in mammals [22], and in few other proteins, such as sodium voltage-gated channel alpha subunit 5 (SCN5A). Then, to better understand the impairment of host polysialylation in *T. cruzi* infection, we investigated whether NCAM1 and SCN5A would be differentially modulated in hiPSC-CM and SH-SY5Y during infection. In hiPSC-CM, NCAM1 levels did not differ between uninfected or infected cells (**Fig 1F and 1H**), in accordance with the observation that this cultured-cells do not express PSA-NCAM, the polysialylated form of NCAM1. On the other hand, the abundance of polySia in SCN5A, the primary cardiac isoform of the voltage-gated sodium channel (Na$_v$ 1.5), was significantly reduced in hiPSC-CM- infected cells (**Fig 1G and 1H**). Interestingly, it was shown that differential sialylation in the primary cardiac isoform of Na$_v$ 1.5 channel generates more depolarized potentials in atrial and ventricular cardiomyocytes, suggesting that cardiac contractility may be modulated by changes in sialic acids associated with the channels [51,67]. Furthermore, in *ST8Sia2* knockout mice, polysialylation deficiency compromises action potential and Nav functions in atrial and ventricular cardiomyocytes [29,30]. Altogether, these results support our hypothesis that *T. cruzi* infection manipulates the polysialylation machinery of the host, compromising

the expression of ST8Sia2, polySia, and SCN5A, which are targets of polysialylation, and thus may alter the complex biology of cardiomyocytes.

Similar to our findings with *T. cruzi*-infected hiPSC-CM, we detected reduced levels of ST8Sia2 and polySia in *T. cruzi*-infected human neuroblastoma (SH-SY5Y) cells, in addition to lower polysialylation of PSA-NCAM, expressed by this cell line (**Fig 2G and 2H**), or SCN5A (**Fig 2E and 2H**). Metabolic differences between the cell lines employed may explain the lower levels in the polysialylated SCN5A and NCAM1, as well as in ST8Sia2 expression observed in SH-SY5Y in relation to hiPSC-CM cells (**Figs 1 and 2**). Nevertheless, both results reinforce our hypothesis that the polysialylation machinery is modulated during *T. cruzi* infection. Although two different polysialyltransferases (ST8Sia2 and ST8Sia4) are involved in the synthesis of polySia chains, it was previously demonstrated that SH-SY5Y expresses high levels of *ST8Sia2* mRNA, but it is almost devoid of *ST8Sia4* mRNA [68] and expresses polySia bound to the two transmembrane proteoforms of NCAM, NCAM140, and NCAM180. Indeed, all expressed NCAM proteoforms are polysialylated [69], and at least, ST8Sia2 is related to polySia modulation of glycoproteins in cells infected by *T. cruzi*. Previous studies have demonstrated that post-translational modifications (PTMs) can profoundly affect protein stability and expression [70,71]. Consequently, we hypothesize that decreased levels of polysialylation may compromise the stability of NCAM protein and, as a result, its expression.

Although the role of NCAM is relatively little explored in the heart, its expression is known to be differentially regulated during the cardiac system development [72]. In the fetal heart, NCAM is expressed in the myocardium and ventricular conduction system, but in the adult heart, its expression is restricted to the ventricular conduction system [73]. Additionally, NCAM1 deletion compromises the expression of genes essential for the coordination of rhythmic contractions of the conduction system in a subpopulation of ventricular cardiomyocytes [29]. Considering that in chronic Chagas cardiomyopathy, the blockade of the ventricular conduction system by the high and persistent parasitism in the cardiac tissue leads to ventricular dysfunction, which can result in heart failure [34, 74, 75], it is tempting to speculate that somehow, the dysfunction in the ventricular conduction system frequently seen in chronic chagasic cardiomyopathy may result partly from manipulation of the polysialylation machinery that compromises PSA-NCAM and/or SCN5A expression.

Interestingly, both chemical or genetic inhibition of ST8Sia2 by CMP or siRNA *ST8Sia2*, respectively, increase the number of intracellular parasites in SH-SY5Y cells measured 48 h.p.i (**Fig 5**). This fact strongly suggests that during the *T. cruzi* infection, modulation of host polysialylation may be a strategy adopted by the parasite to ensure success in establishing the infection, either by affecting the invasion step (e.g. by decreasing the repulsive negative charges of surface molecules with time) or by affecting the intracellular cycle of the parasite. However, the mechanism, as yet remains elusive. Since the highest changes in the levels of ST8Sia2 were observed from 24 h.p.i. (**S1B–S1E Fig**). In this context, it is interesting to note that both, quantity and quality, of polySia synthetized by ST8SIA2 or ST8SIA4 are similar, but the resulted polySia-NCAMs exhibit different attractive and repulsive properties, which may affect biological functions [76]. To date, no study has explored the possible involvement of polysialylation in *T. cruzi* infection. However, previous studies reported that in Chagas disease, possible disorders of aberrant glycosylation are due to modifications in the sialylation of glycoproteins. As demonstrated before by Libby et al in 1986, *T. cruzi* trypomastigotes modify the surface of myocardial and vascular endothelial cells by desialylation [45].

In the present study, we demonstrate that *T. cruzi* infection leads to downregulation of *ST8Sia2*, compromising the levels of NCAM1 and SCN5A, both molecules previously reported as polysialylated, thereby providing a favorable environment for intracellular parasite replication and survival. To date, a possible mechanism explaining the modulation of host

polysialylation during *T. cruzi* infection has not been demonstrated. Here, we show that the silencing of *ST8Sia2* and disruption of host polysialylation modulate oxidative stress in SH-SY5Y cells (**Fig 6**).

A previous study reported that the removal of sialic acids in human endothelial cells modulates signaling mediated by Nuclear Factor E2-related factor 2 (Nrf2), promoting mitochondrial ROS accumulation [77]. Goswmami and Konet [78] experimentally demonstrated that the induction of oxidative stress in human platelets causes desialylation of platelet glycoproteins. Sialic acid has the ability to protect against oxidative damage by inducing a significant increase in the levels of antioxidant components, such as superoxide dismutase (SOD) and glutathione peroxidase (GSH-Px), thereby reducing the ROS production [79].

It has been demonstrated that the endogenous sialidase Neuraminidase 1 (NEU1) can cleave polySia [80]. The reduction of NEU1 increases the expression of target genes associated with Nrf2-mediated signaling, including Heme oxygenase-1 (HO-1), directly implicating sialic acid in the regulation of Nrf2 signaling, a pathway involved in maintaining the redox homeostasis [77]. Furthermore, it has been demonstrated that the overexpression of HO-1 reduces *T. cruzi* parasitism both *in vitro* and *in vivo* infection [81]. As a result, higher levels of NEU1 may reduce polySia levels, increase oxidative stress, and promote parasite survival. Considering the impact of host cell polysialylation on *T. cruzi* infection presented here and the existence of a possible interplay between the downregulation of polysialylation and the modulation of the stress response, a proteomic approach was used to identify differentially regulated proteins involved in the stress response pathway in SH-SY5Y si*ST8Sia2* cells (*ST8Sia2* gene silenced) infected by *T. cruzi* (**Fig 7**).

Our data demonstrated the upregulation of PLP2 and S100A6 in *T. cruzi*-infected si*ST8Sia2* SH-SY5Y cells. It has been previously demonstrated that the modulation of PLP2 levels in mice and human fibroblasts favors stress response [82]. Upregulation of S100A6 has been shown in Hep-2 cells, an epithelial human cell line, under stress conditions [83]. Another study reported that the increase in S100A6 expression, combined with its ligands such as heat shock proteins, modulates the cellular stress response [84]. Additionally, Zhou *et al.* [85] demonstrated that overexpression of *S100A6* in host cells favors infection by *Toxoplasma gondii*, the etiological agent of toxoplasmosis (PMID: 34950858).

Proteomic analysis of *T. cruzi*-infected si*ST8Sia2* SH-SY5Y cells revealed the downregulation expression of PAK2 and MSH2 proteins. Supporting our hypotheses, previous studies have shown that under oxidative stress conditions, reduced levels of PAK2 were detected in cardiomyocytes [86]. In the same line, it was demonstrated that the depletion of PAK2 in cardiomyocytes compromises the establishment of a redox response under stress conditions [87], reinforcing that the reduced levels of PAK2 favors the generation of oxidative stress [88] The interplay between reduced MSH2 levels and the oxidative stress response has also been reported in human cells [89]. Due to this, our findings point towards a correlation between diminished levels of ST8Sia2 and increased susceptibility of host cells to induce oxidative stress, leading to elevated levels of ROS that could create an advantageous milieu for *T. cruzi* parasite replication.

In summary, the present study demonstrates the impairment of the host polysialylation machinery during infection by *T. cruzi*. We have shown that *T. cruzi* infection promotes the differential regulation of ST8Sia2 and polySia in host cells (**Fig 8**) and suggests a possible involvement of oxidative stress response during infection. Using different experimental approaches, we also showed a reduction in NCAM and SCN5A, both targets of polysialylation, which strongly suggests that ventricular conduction dysfunction frequently observed in chronic Chagas cardiomyopathy may be due to the remodeling of the polysialylation of these molecules.

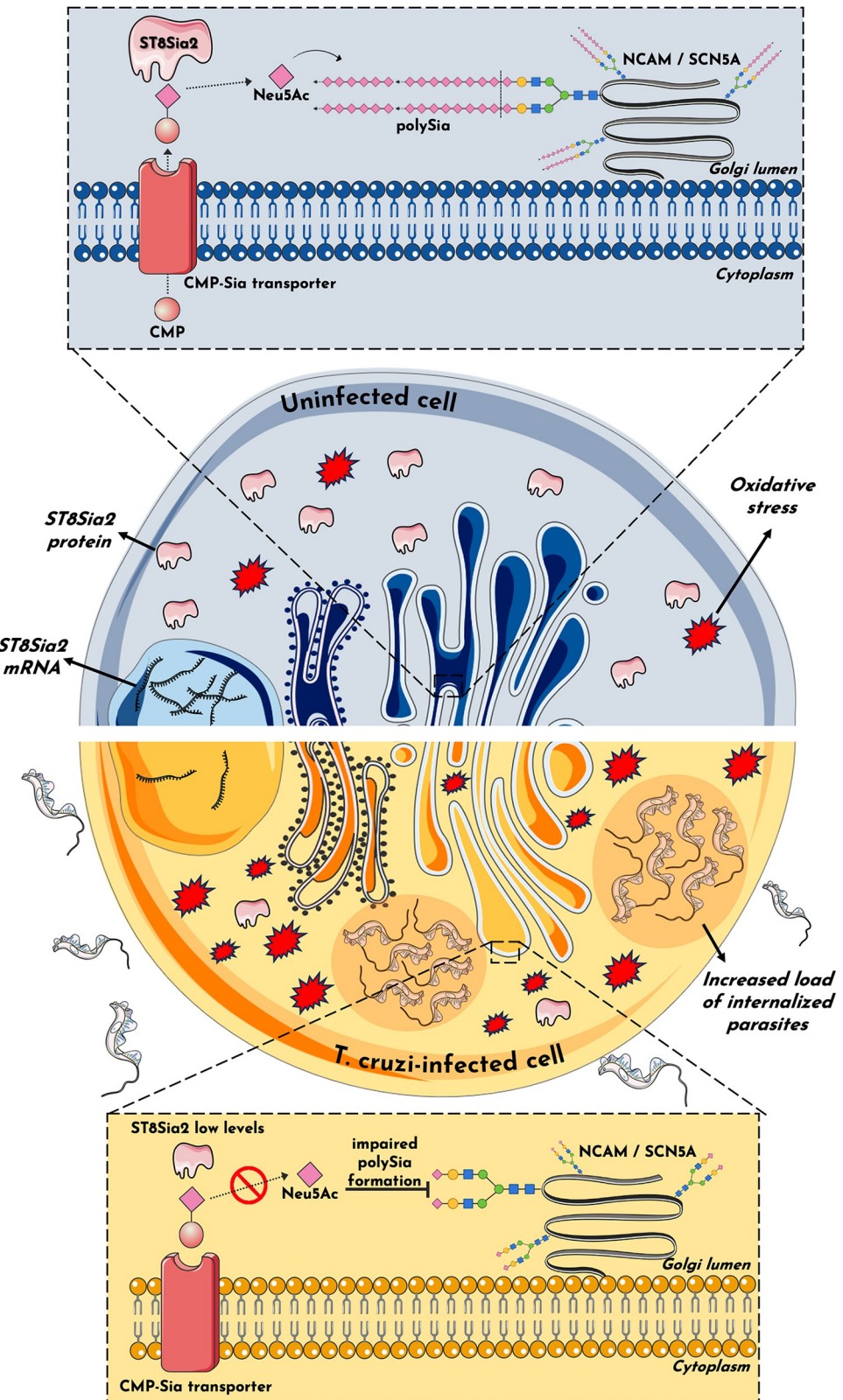

**Fig 8. Potential mechanism of host polysialylation modulation during *T. cruzi* infection.** The findings in this study suggest an impact on host polysialylation during *T. cruzi* infection. Under conditions of homeostasis, ST8Sia2 catalyzes the transfer of Neu5Ac monomers from cytidine monophosphate N-acetylneuraminic acid (CMP-Neu5Ac) to

glycoproteins within the Golgi complex, forming a polymer of α-2,8-glycosidic linkages between Neu5Ac monomers known as polySia. This phenomenon, termed polysialylation, assumes pivotal roles across a spectrum of cellular functions. The addition of polySia into molecules such as NCAM1 and SCN5A influences cardiac contractility and neural communication, exemplifying its multifaceted significance. In this study, we show that host polysialylation impact *T. cruzi* infection dynamics. Our investigation has unveiled that *T. cruzi* infection diminishes the expression/ abundance of the ST8Sia2 enzyme, compromising polySia formation, and its subsequent addition to polysialated target molecules, including SCN5A and NCAM1. Consequently, the attenuation of ST8Sia2 and polySia levels may influence the host cell susceptibility to generate an oxidative stress environment that may favor *T. cruzi* infection, as polysialylation modulation is accompanied by an increase in the number of internalized parasites and susceptibility to oxidative stress. It is plausible to posit that the modulation of host polysialylation by the parasite serves as a determinant in the pathogenesis of Chagas disease. Parts of the figure were drawn by using pictures from Servier Medical Art. Servier Medical Art by Servier is licensed under a Creative Commons Attribution 3.0 Unported License (https://creativecommons.org/licenses/by/3.0/).

Furthermore, our findings show that reduced levels of *ST8Sia2* heighten host cells susceptibility to oxidative stress upon *T. cruzi* infection. This suggests that downregulation of *ST8Sia2* disrupts polySia production, diminishing polysialylation and enhancing ROS generation, potentially fostering a conducive environment to *T. cruzi* parasite replication. Importantly, pharmacological and genetic inhibition of ST8Sia2 in infected cells demonstrated an increase in intracellular parasite numbers, highlighting a possible functional role for manipulating polysialylation during *T. cruzi* infection. Of note, our study does not provide a comprehensive mechanistic understanding, but will help inform future studies aimed at elucidating the functions of polySia in infectious diseases, with a focus on *T. cruzi* infection and Chagas disease pathogenesis. Furthermore, studying different Discrete Typing Units (DTUs) can provide a comprehensive view on the role of ST8Sia2 in strain-specific pathogenic mechanisms of *T. cruzi*.

## Materials and methods

### Public RNA-seq data reanalysis

The fastq files were downloaded from the https://sra-explorer.info/ platform with the BioProject accession number PRJNA532430 [49]. The 'FastQC' module was used to report the quality reads, followed by the trimmed of adapters using TrimGalore! set to the single-end library. The TrimGalore! output sequences were aligned to the human reference genome hg38 using the HISAT2 platform [90]. A count table was generated using the FeatureCounts [91] algorithm. The differently regulated genes were analyzed by the limma, Glimma, edgeR, and Homo.sapiens packages applying a cut-off of |log2FC|>1 and a p-adjusted value <0.05 (Benjamini-Hochberg). Statistical data of the glycogenes of interest were filtered and the chord_dat() function, available in the GOplot package, was used to plot the chord plot. Volcano plot and heatmap graphics were made using Perseus software.

### Human cell culture

Human induced-pluripotent stem cells (hiPSC) were differentiated into Human induced-pluripotent stem cells derived cardiomyocyte (hiPSC-CM) on Geltrex (Thermo Fischer, Waltham, MA, USA) coated plates in Cardiomyocytes-Plating Medium (Pluricell Biotech), using an established differentiation protocol [92]. All experiments were carried out between day 30–35 of differentiation. The human neuroblastoma SH-SY5Y cells (ATCC-CRL-2266) were cultured in Dulbecco's Modified Eagle Medium/Nutrient Mixture F-12 (DMEM/F12) (Gibco, Life Technologies, Carlsbad, CA, USA), supplemented with 10% heat-inactivated fetal bovine serum, and 1% penicillin/streptomycin. All cells line were maintained in a humidified 5% $CO_2$ atmosphere at 37˚C.

### *T. cruzi* culture and infection assay

*T. cruzi* trypomastigote (Y strain) was routinely maintained by infection in monkey kidney epithelial cells (LLC-MK2) in Roswell Park Memorial Institute 1640 Medium (RPMI) supplemented with 2% fetal bovine serum in a humidified 5% $CO_2$ atmosphere at 37˚C, according to previously described experimental procedures [93]. Five days after infection, the trypomastigotes released in the supernatants were purified using a DEAE-cellulose chromatography column. Therefore, the cells were washed in phosphate-buffered saline (PBS) by centrifugation at $10.000 \times g$ for 12 min, and resuspended in adequate culture medium for each human cell line used. The total number of parasites/ml was counted in a Neubauer chamber. hiPSC-CM and/or SH-SY5Y cells were seeded at a density of $5x10^5$ cells/well in 24-well or 6-well microplates and infected with purified trypomastigotes at the rate of 5:1 (trypomastigotes:hiPSC-CM) in a volume of 500 µL per well [49]. After 3 h of incubation, the media was replaced with fresh media to remove the non-internalized parasites. The impact of *T. cruzi* infection on host polysialylation was evaluated at 48 h post-infection (h.p.i). In all experiments, an uninfected group was subjected to the same experimental conditions, except for incubation with *T. cruzi*.

### Immunofluorescence assay

hiPSC-CM or *T. cruzi*-infected SH-SY5Y cells distributed in round glass coverslips (13 mm) contained in 24-well microplates with flat bottom were fixed with PBS/4% paraformaldehyde for 15 min at room temperature (RT) and permeabilized with 0.1% Triton X-100 (Sigma, Chemical Company, St. Louis, MO, USA) for 15 min at RT. For evaluation the efficiency of differentiation on hiPSC-CM, coverslips were blocked with PBS/2% Bovine Serum Albumin (BSA) solution, and incubated with primary antibodies: mouse anti-cardiac troponin T (HyTest clone 4T19/2) and rabbit anti-α-actinin (Millipore) for 30 min at RT. After washing four times with PBS, the cells were incubated for 1 h at RT with the secondary antibodies: goat anti-rabbit immunoglobulin G (IgG) Alexa Fluor 488 and/or goat anti-mouse IgG Alexa Fluor 594 (Thermo Fisher Scientific, Waltham, Massachusetts, USA). The coverslips were washed four times and prepared for visualization by adding one drop of mounting medium (Slow Fade Gold Antifade; Thermo Fisher Scientific) with 4',6-diamidino-2- phenylindole (DAPI). For evaluation of the ST8Sia2 and polySia levels in *T. cruzi*-infected SH-SY5Y cells, the cells seeded on coverslips, fixed and permeabilized were blocked with PBS/2% Bovine Serum Albumin (BSA) solution, and incubated with primary antibodies: ST8Sia2 monoclonal antibody (Catalog WH0008128M1, Sigma-Aldrich, St. Louis, MO, USA) and polySia-specific monoclonal antibody (mouse IgG2a, clone 735) for 30 min at RT. After washing four times with PBS, the cells were incubated for 1 h at RT with the secondary antibodies: Alexa Fluor 488 goat anti-mouse IgG (Thermo Fisher Scientific, Waltham, Massachusetts, USA) or Alexa Fluor 594 goat anti-mouse IgG (Thermo Fisher Scientific, Waltham, Massachusetts, USA). The coverslips were washed four times and prepared for visualization by adding one drop of mounting medium (Slow Fade Gold Antifade; Thermo Fisher Scientific) with 4',6-diamidino-2- phenylindole (DAPI). To determine the number of parasites internalized in SH-SY5Y cells, the cells seeded on coverslips, fixed and permeabilized as above, were prepared for visualization by adding a drop of mounting medium (Slow Fade Gold Antifade; Thermo Fisher Scientific) containing 4',6-diamidino-2-phenylindole (DAPI), used for imaging, which was performed using a Conventional Fluorescence Microscope (Zeiss). polySia-specific monoclonal antibody (mouse IgG2a, clone 735) were kindly provided by Dr. Martina Mühlenhoff (Institute of Clinical Biochemistry, Hannover Medical School, Germany).

## Quantification of immunofluorescence

Fluorescence microscopy images were analyzed by ImageJ software, version 1.49v (Rasband, W.S., ImageJ, U. S. National Institutes of Health, Bethesda, Maryland, USA, http://imagej.nih. gov/ij/) to evaluate ST8Sia2 and polySia-stained host cells. Using the calculation for corrected total cell fluorescence (CTCF) = integrated density–(area of selected cell × mean fluorescence of background readings), as described by Gachet-Castro et al. [94]. The fluorescence intensity of each cell was calculated. Subsequently, graphs were plotted with the fluorescence average values using GraphPad Prism 9.0 program (GraphPad Software, Inc., San Diego, CA).

## Total cell lysates

hiPSC-CM and SH-SY5Y cells were infected with *T. cruzi* trypomastigotes at the rate of 5:1 (trypomastigotes:hiPSC-CM) for 48 h. After 48 h.p.i, the cells were washed four times with PBS to remove extracellular parasites, and lysed with radioimmunoprecipitation assay (RIPA) buffer [20 mM Tris pH 7.2, 150 mM NaCl, 1% Triton X-100, 1% sodium deoxycholate, 0.1% sodium dodecyl sulphate (SDS)] containing a cocktail of protease (cOmplete, Sigma-Aldrich, St. Louis, MO, USA) and phosphatase (PhosStop, Sigma-Aldrich) inhibitors. The cell lysates were kept on ice for 10 min, centrifuged at 14.000 x g for 10 min at 4˚C to pellet cell debris. Supernatants were collected, and the total protein was measured using the Kit Pierce BCA Protein Assay (Thermo Scientific) according to the manufacturer's instructions. In all experiments, an uninfected group was subjected to the same experimental conditions. All Samples were resuspended in sample buffer for analysis by SDS-PAGE. The protein extract from human rhabdomyosarcoma cell line TE671 in sample buffer [2% sodium dodecyl sulfate, 10% glycerol, 0.002% bromophenol blue, 0.0625 M Tris-HCl pH 6.8, 5% 2-mercaptoethanol] were gifts from Dr. Martina Mühlenhoff (Institute of Clinical Biochemistry, Hannover Medical School, Germany).

## Immunoblotting assay

Proteins extracted from total cell lysate (15μg of proteins) were resolved by SDS-PAGE and transferred to PVDF membranes, which were directly incubated with blocking buffer (5% BSA in Tris-buffered saline (TBS) at 0.05% Tween-20 (TBS/T) for 1 h. Subsequently, membranes were washed three times with TBS/T solution and incubated *overnight* with the primary antibodies: anti-ST8Sia2, anti-polySia, anti-NCAM1, anti-PSA-NCAM, anti-SCN5A, and Anti-ACTB, and washed three times with TBS/T. Then, the membrane was incubated with the respective secondary antibodies for 1 h at RT, followed by the incubation with a secondary antibody for 1h at room temperature. The immunoreactive bands were detected with the ChemiDoc XRS Imaging System equipment and protein quantification was performed using the ImageJ software. polySia-specific monoclonal antibody (mouse IgG2a, clone 735) and NCAM1-specific monoclonal antibody (mouse IgG1, clone 123C3) were kindly provided by Dr. Martina Mühlenhoff (Institute of Clinical Biochemistry, Hannover Medical School, Germany). All antibodies used in this study were tested for cross-reactivity against *T. cruzi* trypomastigote cell lysates.

## Drugs treatment and chemical ST8Sia2 inhibition

SH-SY5Y cells were pretreated with 0.5 mM cytidine 5'-monophosphate (CMP, Sigma-Aldrich) or culture medium (negative control) 24 h before infection and 24 h post-infection with *T. cruzi* to evaluate the impact of pharmacological inhibition of ST8Sia2 activity during infection and replication of the parasite. After incubation with CMP for 24 h, the CMP-

containing medium was replaced with fresh medium, and the cells were infected with *T. cruzi* trypomastigotes in a 5:1 ratio (trypomastigotes:hiPSC-CM). 24 h post-infection, the treatment with 0.5 mM CMP was repeated for another 24 h. Guanosine 5′-monophosphate (GMP), with no effect on ST8Sia2 inhibition, was used as a control. In all experiments, a group of infected cells (negative control of ST8Sia2 inhibition) was subjected to the same experimental conditions, except for incubation with pharmacological inhibitor.

## siRNA-Directed Inhibition of *ST8Sia2*

Predesigned siRNAs for the *ST8Sia2* transcripts (si*ST8Sia2*) (hs.Ri.ST8Sia2.13.2; hs.Ri. ST8Sia2.13.3) were purchased from Integrated DNA Technologies (IDT, Coralville, IA). Transfection was performed using Lipofectamine RNAiMAX (Thermo Fisher Scientific, Waltham, MA, USA). Lipofectamine RNAiMAX:siST8Sia2 Mix (0.6 μL Lipofectamine:100 nM siRNAs) was formed in Opti-MEM (Thermo Fisher Scientific, Waltham, MA, USA) at room temperature for 20 min and added to SH-SY5Y cells in antibiotic-free medium for overnight transfection at 37˚C and 5% $CO_2$. After incubation, fresh Dulbecco's Modified Eagle Medium/ Nutrient Mixture F-12 (DMEM/F12) (Gibco®, Life Technologies, Carlsbad, CA, USA), supplemented with 2% heat-inactivated fetal bovine serum was added, and the cells were infected with *T. cruzi* trypomastigotes in a 5:1 ratio (trypomastigotes:hiPSC-CM). 24 h post-infection, *T. cruzi*-infected SH-SY5Y cells were transfected again with Lipofectamine RNAiMAX:siST8-Sia2 Mix (0.6 μL Lipofectamine:100 nM siRNAs) in Opti-MEM (Thermo Fisher Scientific, Waltham, MA, USA) antibiotic-free medium for an additional 24 h, at 37˚C and 5% $CO_2$. siRNA negative control (NTC) and medium alone (Medium) were used as negative controls. Transfection efficiency was evaluated by the relative expression of the *ST8Sia2* using Quantitative real-time reverse transcription PCR (qRT-PCR).

## Quantitative real-time reverse transcription PCR (qRT-PCR)

hiPSC-CM were infected with *T. cruzi* trypomastigotes (Y strain) at the rate of 10:1 or 5:1 (trypomastigotes:host cell). Total RNA from these samples was extracted using TRIzol according to the manufacturer's instructions, and the RNA was converted into cDNA using the High-Capacity cDNA Reverse Transcription kit (Applied Biosystems). qRT-PCR was performed in a final volume of 10 μL using Maxima SYBR Green qPCR Master Mix (Thermo Scientific) and the quantification was done by QuantStudio 3 Real-Time PCR System (Applied Biosystems). The cycling conditions were as follows: initial denaturation at 95˚C for 10 min, followed by 40 cycles of denaturation at 95˚C for 15 s and annealing/extension at 60˚C for 60 s. The relative expression of the transcript was quantified using the ΔΔCt method [95] and normalized to β-actin expression. The specificity of amplification was determined using melting curve analysis. The following PCR primers were used: *β-actin* (F- CATGTACGTTGCTATCCAGGC, R-CTCCTTAATGTCACGCACGAT), and *ST8Sia2* (F- CGTTCGCGGATACTGGCT, R-GCCAGAAGCCGTAGAGGTAG).

## MTT assay

SH-SY5Y cells were seeded in 96-well microplates (1 × 105 cells/well) and treated with cytidine 5'-monophosphate (CMP; 62.5–500 μM) or siRNA *ST8Sia2* (50–150 nM) for 24 and/or 48 h at 37˚C. Guanosine 5'-monophosphate (GMP), siRNA negative control (NTC), and/or medium alone were used as negative controls for cell death. The cell viability of the cells was determined after the reduction of MTT (Sigma-Aldrich) to produce formazan crystals [96]. The procedure was performed as described by Barboza *et al.* [3]. Mitochondrial activity was expressed as a percentage after comparison with the absorbance of unstimulated SH-SY5Y cells.

## Quantification of polysialic acid

For quantification of polysialic acid (polySia), SH-SY5Y cells infected with *T. cruzi*, as well uninfected cells, were incubated with iEndo-N (inactive Endo-N-acetylneuraminidase) for 1h at 37°C. In some experiments, polySia quantification was performed in SH-SY5Y cells with silenced *ST8Siaa2* (*siST8Sia2*). After the incubation time, 30 μg of protein from each sample were immobilized on PVDF membrane and submitted to direct hydrolysis of the sialic acid with 2M acetic acid for two hours at 80° C. Hydrolyzed Sia were derivatized with 1,2-Diamino-4,5-methylenedioxybenzene (DMB; P/N D4784) obtained from Sigma-Aldrich and separated in reversed-phase chromatography with fluorescence detection using a Vanquish UPLC (Thermo Fisher). iEndo-N was a gift from Dr. Martina Mühlenhoff (Institute of Clinical Biochemistry, Hannover Medical School, Germany).

## Measurement of cytosolic and mitochondrial ROS levels

Cytosolic and mitochondrial ROS levels were quantified using flow cytometry with the probes: dihydroethidium (DHE) (Sigma-Aldrich) and MitoSox Red (Thermo Fisher Scientific), respectively, as previously described [97]. SH-SY5Y si*ST8Sia2* cells plated in 6-well microplates ($1 \times 10^6$ cells/well) were treated or untreated with antimycin A (50 μM) or N-acetyl cysteine (5mM) for 4 hours to induce or inhibit oxidative stress, respectively. Negative control siRNA (NTC) was used as a negative control for *ST8Sia2* silencing. Lipofectamine RNAiMAX was added in all controls used in experiments with siRNA. Subsequently, cells were washed three times with phosphate-buffered saline (PBS) and incubated with DHE for 40 minutes or with 5 μM of MitoSox Red for 10 minutes at 37°C and 5% $CO_2$. After incubation, cells were washed three times with PBS and centrifuged at 300 x g for 7 min at 26°C, and then transferred to cytometry tubes. Fluorescence intensity was analyzed on the LSR Fortessa cell analyzer flow cytometer (BD Biosciences) with excitation at 488nm and emission at 620nm. A minimum of 500.000 events was collected. Six experiments were performed for statistical analysis.

## Measurement of Mitochondrial Calcium Ion ($Ca^{2+}$) Concentration

Changes in mitochondrial $Ca^{2+}$ ion were measured using the Rhod-2 probe (Thermo Fisher Scientific, Waltham, MA, USA). SH-SY5Y si*ST8Sia2* cells plated in 6-well microplates ($1 \times 10^6$ cells/well) were incubated with 5 μM Rhod-2 plus pluronic acid F-127 (5 μM; Sigma-Aldrich, St. Louis, MO, USA) and bovine serum albumin (30μg/mL; Sigma-Aldrich) at 37°C and 5% $CO_2$ for 60 minutes, as previously described [97]. Negative control siRNA (NTC) was used as a negative control for *ST8Sia2* silencing. Lipofectamine RNAiMAX was added in all controls used in experiments with siRNA. After incubation, cells were washed three times with PBS, centrifuged at 300 x g for 7 min at 26°C, and then transferred to cytometry tubes. Maximum fluorescence was measured using tubes with cells incubated with ionomycin (1 μM) for 2 minutes, and minimum fluorescence was measured using tubes with cells incubated with EDTA (8mM) for 2 minutes. Fluorescence intensity was analyzed on the LSR Fortessa cell analyzer flow cytometer (BD Biosciences) with excitation at 395nm and emission at 525nm. A minimum of 500.000 events was collected. Next, mitochondrial $Ca^{2+}$ concentration was calculated with the following equation: $Ca^{2+}$ mitochondrial = Kd (Fsample—Fmin) / Fmax—Fsample, where F represents the fluorescence value and Kd was 390nM. Six experiments were performed for statistical analysis.

## Determination of Mitochondrial Transmembrane Potential (ΔΨm)

Changes in ΔΨm were measured using the MitoTracker Red CMXRos probe (Thermo Fisher Scientific). SH-SY5Y si*ST8Sia2* cells plated in 6-well microplates (1 x 106 cells/well) were

incubated with 100nM MitoTracker Red CMXRos at 37˚C with 5% CO2 for 30 minutes, as previously described [97]. Negative control siRNA (NTC) was used as a negative control for *ST8Sia2* silencing. Lipofectamine RNAiMAX was added in all controls used in experiments with siRNA. After incubation, cells were washed three times with PBS and centrifuged at 300 x g for 7 min at 26˚C, and then transferred to cytometry tubes. Fluorescence intensity was analyzed on the LSR Fortessa cell analyzer flow cytometer (BD Biosciences) with excitation at 488 nm and emission at 620nm. A minimum of 500.000 events was collected. Six experiments were performed for statistical analysis.

## Sample preparation for Mass-Spectrometry Based-Proteomics

*T. cruzi*-infected *ST8Sia2* SH-SY5Y cells (*siST8Sia2*) were lysed in 8M urea containing protease using a sonicator for three cycles at 10% amplitude. *T. cruzi*-infected SH-SY5Y cells treated with culture medium alone (Medium) were used as negative control for *ST8Sia2* silencing. Following extraction, a 1 μL aliquot of each sample was taken for protein quantification using the Qubit method. We used 30 μg of each sample for the digestion protocol. Ammonium bicarbonate (Ambic) was added to a final concentration of 50 mM, and the proteins were reduced with 10 mM DTT (DL-Dithiothreitol, Sigma-Aldrich) for 45 min at 30˚C, alkylated with 40 mM iodoacetamide (Sigma-Aldrich) for 30 min at room temperature in the dark. The solution was diluted with 50 mM Ambic to 1M Urea, and the proteins were digested with trypsin at a ratio of 1:50 (μg trypsin/μg protein) for 16 hours at 30˚C. The reaction was stopped with 1% formic acid (pH < 2) and desalted using C18 columns (StageTips).

## Nano-LC-MS/MS Mass spectrometry analysis

The peptides were analyzed by nano-high performance liquid chromatography coupled with tandem mass spectrometry (nano-LC-MS/MS) using a nanoLC Easy-LTQ Orbitrap Velos-ETD (Thermo Scientific). The nanoLC gradient was from 0% to 30% solvent B (solvent A: 0.1% formic acid; solvent B: 90% acetonitrile, 0.1% formic acid) over 80 minutes at a flow rate of 300 nL/min. Precursor ions were selected based on data-dependent acquisition in positive mode. The 400–1600 m/z MS1 scans were recorded in the Orbitrap mass analyzer at a resolution of 120,000, with a target AGC of $3 \times 10^6$ and a maximum injection time of 100 ms. The MS/MS spectra of the 20 most abundant precursor ions per scan were obtained through collision-induced dissociation (CID) with Orbitrap acquisition at a resolution of 30,000. Precursor ions were selected using an isolation window of 1.2 Da and fragmented by CID, with a maximum injection time of 150 ms.

## Database search and proteomics data analysis

Raw data were searched with Proteome Discoverer software (version 3.0) using the Sequest search engine. The Swiss-Prot Homo sapiens database was concatenated with the unreviewed *Trypanosoma cruzi* database downloaded in March 2024. Trypsin was used as the cleavage enzyme and the search was limited to full cleavage; methionine oxidation and cysteine carbamidomethylation were defined as variable and fixed modifications, respectively. Other search criteria included: 2 missed cleavages; mass tolerance of 20 ppm (precursor ions), 0.5 Da (fragment ions), and minimum peptide length of 7 amino acids. The False Discovery Rate (FDR) was set to 1% for identifying peptide spectrum matches (PSMs), peptides, and proteins. Protein quantification was performed using the label-free quantification module. Functions for data transformation and filtering were applied using Perseus software. Proteins quantified in all biological replicates (out of a total of three replicates per group) were considered for differential regulation analysis (p-value < 0.05) by applying a t-test. Proteins considered exclusive

were quantified in the three biological replicates of a group and none or one biological replicate of the group in the comparison. The chord plot was built by the GOplot package (10.1093/bioinformatics/btv300), available in Bioconductor. The mass spectrometry proteomics data have been deposited to the ProteomeXchange Consortium via the PRIDE [98] partner repository with the dataset identifier PXD053813 (Reviewer account details: Username: reviewer_pxd053813@ebi.ac.uk | Password: ULuzdFYhDYMe).

## Statistical analysis

Initially, the results were tested for normal distribution and homogeneous variance. Once the normal distribution and homogeneous variance was confirmed, the One-way ANOVA test was applied. In the absence of normal distribution and heterogeneous variance, the Kruskal-Wallis test was applied. The statistical evaluation of the differences between the means of the groups were determined by the one-way analysis of variance, followed by the Bonferroni multiple comparison test and, in the non-parametric test, the comparisons between the groups were carried out by the Dunn's multiple comparison test. For all statistical analyzes, the Prism 9.0 program (GraphPad Software, La Jolla, CA, USA) was used.

## Supporting information

**S1 Fig. *T. cruzi* infection modulates genes involved in host N-glycosylation machinery.** Bioinformatics analysis of Human induced pluripotent stem cell-derived cardiomyocytes (hiPSC-CM) infected with *Trypanosoma cruzi* (Y strain). **(A)** Chordplot indicating the modulation of N-glycosylation machinery transcripts at 24 h (inner), 48 h (middle), and 72 h (outer) post-infection (h.p.i) with *T. cruzi* (infected vs non-infected; INFvsUNF). The R GOplot package was used to build chord plot. Red and blue colors indicate upregulated and downregulated transcripts, respectively; **(B-E)** Volcano plot (left graph) and Violin plot (right graph) of transcripts encoding enzymes involved in the N-glycosylation pathway that were differentially regulated in *T. cruzi*-infected hiPSC-CM, focusing on the modulation of the *ST8Sia2* transcripts after 0 h **(B)**, 24 h **(C)**, 48 h **(D)**, and 72h **(E)** post-infection. The logFC (INFvsUNF) indicates transcripts upregulated in orange and downregulated in dark blue.
(TIF)

**S2 Fig. Human pluripotent stem cells (hiPSC) are efficiently differentiated into cardiomyocytes (hiPSC-CM), and do not express PSA-NCAM. (A)** Representative immunofluorescence images of hiPSC-CM confirming the presence of α-actinin and troponin T (sarcomeric proteins). Cell nucleus were stained with DAPI (scale bars = 10 μm). **(B)** Histogram of Troponin T (TNNT2+) staining the differentiation of hiPSC into hiPSC-CM. **(C)** Representative western blot images of PSA-NCAM levels in uninfected hiPSC-CM (lane 1), *T. cruzi*-infected hiPSC-CM (lane 2), TE671 cells (lane 3), and *T. cruzi* trypomastigote (lane 4). hiPSC-CM cells were seeded in 6-well microplates (1 x $10^6$ cells/well) and infected with *T. cruzi* trypomastigotes (Y strain) in a ratio of 1:5 (SH-SY5Y:trypomastigotes) for 48 h at 37˚ C. Protein extracts from TE671 cell lysates (in Laemmli buffer) and *T. cruzi* trypomastigote (in RIPA buffer) were used as a positive and negative control for PSA-NCAM, respectively. 15 μg of protein extracts were used to analyze the PSA-NCAM levels by Western blotting using a specific anti-PSA-NCAM antibody.
(TIF)

**S3 Fig. Chemical and genetic inhibition of ST8Sia2 in SH-SY5Y cells does not affect cell viability. (A,B)** SH-SY5Y cells were seeded in 96-well microplates (1 × $10^5$ cells/well), and treated with cytidine 5'-monophosphate (CMP; 62.5–500 μM) or siRNA *ST8Sia2* (50–150 nM)

for 24 and/or 48 h. Guanosine 5'-monophosphate (GMP), siRNA negative control (NTC), and/or medium alone were used as negative controls for chemical or genetic inhibition of ST8Sia2. In all controls used in experiments with siRNA was added Lipofectamine RNAiMAX. Following 24 and 48 h incubation with CMP **(A)** or 24 h incubation with siRNA *ST8Sia2* **(B)**, 3-(4,5-dimethyl-2-thiazolyl)-2,5-diphenyl-2H-tetrazolium bromide (MTT; 50 μg/mL) was added to the cells, and mitochondrial activity was estimated by MTT reduction and expressed as a percentage calculated from the ratio between the absorbance of stimulated and non-stimulated SH-SY5Y cells; **(C)** Relative expression of *ST8Sia2* mRNA measured by qRT-PCR in SH-SY5Y cells treated with siRNA *ST8Sia2* (*siST8Sia2*), following the experimental workflow presented in **D**. siRNA negative control (NTC) were used as negative control for genetic silencing of ST8Sia2. The Ct values of the target transcripts were normalized to the relative expression of *GAPDH* as endogenous control, and the relative expression of ST8Sia2 transcripts was quantified by the $2^{-\Delta\Delta}$ Ct method. Each bar represents the mean of three independent experiments performed in triplicate; **(D)** Experimental workflow adopted to investigate the effect of genetic inhibition of ST8Sia2 in SH-SY5Y cells using 100 nM of siRNA *ST8Sia2*. SH-SY5Y cells were seeded in 24-well microplates (5 x $10^4$ cells/well), and treated with siRNA *ST8Sia2* [100 nM] for 24h, followed by resting time of 24 h additional. After this resting time, siST8Sia2 SH-SY5Y cells were restimulated with siRNA *ST8Sia2* [100 nM] for an additional 24h. siRNA negative control (NTC) were used as negative control. In all controls used in experiments with siRNA was added Lipofectamine RNAiMAX; **(E)** Representative images of polySia levels in siST8Sia2 SH-SY5Y cells. SH-SY5Y cells were transfected with 100 nM ST8Sia2 siRNA using the same experimental strategy as presented in **D**. Following the completion of the ST8Sia2 silencing protocol, siST8Sia2 SH-SY5Y cells were stained with a specific anti-polySia **(E)** antibody, and the labeling for the target was visualized by immunofluorescence microscopy. siRNA negative control (NTC), and/or medium alone were used as negative controls for genetic inhibition of ST8Sia2.Cell nuclei were stained with DAPI (scale bars = 50 μm); **(F)** polySia levels measured in siST8Sia2 SH-SY5Y cells supernatants. SH-SY5Y cells were transfected with 100 nM ST8Sia2 siRNA using the same experimental strategy as presented in **D**. After the completion of the *ST8Sia2* silencing protocol, si*ST8Sia2* SH-SY5Y cells were treated with EndoN [0.5 μg/ml] for 1h at 37°C, and polySia levels were measured in supernatants using UHPLC. The same procedure was applied to siRNA negative control (NTC), and/or medium alone SH-SY5Y cells. The results are expressed in pg/mL. Significant differences compared to the *siST8Sia2* SH-SY5Y cells with siNTC SH-SY5Y cells are shown by (\*) $p < 0.05$, (\*\*) $p < 0.001$, and (\*\*\*\*) $p < 0.0001$; *ns*: not significant. Parts of the figure were drawn by using pictures from Servier Medical Art. Servier Medical Art by Servier is licensed under a Creative Commons Attribution 3.0 Unported License (https://creativecommons.org/licenses/by/3.0/).
(TIF)

**S1 Table. Total proteins identified and quantified in the *T. cruzi*-infected SH-SY5Y cells with and without *ST8sia2* gene silencing.** SH-SY5Y cells were seeded in 6-well microplates (1 x $10^6$ cells/well), and previously treated with siRNA *ST8Sia2* [100 nM] for 24h before infection, followed by infection with *T. cruzi* trypomastigotes (Y strain) at ratio of 1:5 (SH-SY5Y:trypomastigotes). After 24 h.p.i, *T. cruzi*-infected SH-SY5Y cells were treated with siRNA *ST8Sia2* [100 nM] for an additional 24h. Then, the cells were lysed in 8M urea containing protease inhibitors using a sonicator for three cycles at 10% amplitude, reduced, alkylated and digested. The tryptic peptides were analysed by nanoflow liquid chromatography coupled to mass spectrometry to map the differentially proteomic profile in response to *T. cruzi* infection. A triplicate for each condition was run. The total proteins identified in the two conditions along with

their quantitative values in each replicate.
(XLSX)

**S2 Table. Proteins identified and quantified in all replicates of the two conditions under study, SH-SY5Y with and without *ST8Sia2* gene silencing (si*ST8Sia2*) infected with *T. cruzi*.** The total proteins identified in all replicates of the two conditions along with their quantitative values.
(XLSX)

**S3 Table. Differentially regulated proteins considering all replicates of the two conditions under study, SH-SY5Y with and without *ST8Sia2* gene silencing (si*ST8Sia2*) infected with *T. cruzi*.**
(XLSX)

**S1 Data. Raw images data.**
(PDF)

**S2 Data. Raw quantification data.**
(XLSX)

## Acknowledgments

We thank Celia Ludio Braga at the Department of Biochemistry, Institute of Chemistry, University of São Paulo, Brazil for helping with the parasite cell culture. We thank the Vaccine Development Laboratory, Department of Microbiology, Institute of Biomedical Sciences, University of São Paulo, Brazil, for providing the flow cytometry platform (LSR Fortessa cell analyzer flow cytometer, BD Biosciences).

## Author Contributions

**Conceptualization:** Bruno Rafael Barboza, Giuseppe Palmisano.

**Data curation:** Bruno Rafael Barboza, Janaina Macedo-da-Silva, Claudia Blanes Angeli, Cristiane Moutinho-Melo, Giuseppe Palmisano.

**Formal analysis:** Bruno Rafael Barboza, Janaina Macedo-da-Silva, Lays Adrianne Mendonça Trajano Silva, Vinícius de Morais Gomes, Deivid Martins Santos, Antônio Moreira Marques-Neto, Simon Ngao Mule, Maria Julia Manso Alves, Giuseppe Palmisano.

**Funding acquisition:** Giuseppe Palmisano.

**Investigation:** Bruno Rafael Barboza, Antônio Moreira Marques-Neto, Simon Ngao Mule, Juliana Borsoi, Carolina Borsoi Moraes, Martina Mühlenhoff, Walter Colli, Suely Kazue Nagashi Marie, Lygia da Veiga Pereira, Maria Julia Manso Alves, Giuseppe Palmisano.

**Methodology:** Bruno Rafael Barboza, Janaina Macedo-da-Silva, Lays Adrianne Mendonça Trajano Silva, Vinícius de Morais Gomes, Deivid Martins Santos, Antônio Moreira Marques-Neto, Simon Ngao Mule, Claudia Blanes Angeli, Juliana Borsoi, Carolina Borsoi Moraes, Cristiane Moutinho-Melo, Martina Mühlenhoff, Walter Colli, Suely Kazue Nagashi Marie, Lygia da Veiga Pereira, Maria Julia Manso Alves, Giuseppe Palmisano.

**Project administration:** Giuseppe Palmisano.

**Resources:** Martina Mühlenhoff, Suely Kazue Nagashi Marie, Lygia da Veiga Pereira, Maria Julia Manso Alves, Giuseppe Palmisano.

**Software:** Janaina Macedo-da-Silva.

**Supervision:** Maria Julia Manso Alves, Giuseppe Palmisano.

**Validation:** Bruno Rafael Barboza, Giuseppe Palmisano.

**Visualization:** Bruno Rafael Barboza, Janaina Macedo-da-Silva, Lays Adrianne Mendonça Trajano Silva, Vinícius de Morais Gomes, Deivid Martins Santos, Antônio Moreira Marques-Neto, Simon Ngao Mule, Juliana Borsoi, Carolina Borsoi Moraes, Walter Colli, Suely Kazue Nagashi Marie, Maria Julia Manso Alves, Giuseppe Palmisano.

**Writing – original draft:** Bruno Rafael Barboza, Walter Colli, Maria Julia Manso Alves, Giuseppe Palmisano.

**Writing – review & editing:** Bruno Rafael Barboza, Janaina Macedo-da-Silva, Lays Adrianne Mendonça Trajano Silva, Vinícius de Morais Gomes, Deivid Martins Santos, Antônio Moreira Marques-Neto, Simon Ngao Mule, Juliana Borsoi, Carolina Borsoi Moraes, Martina Mühlenhoff, Walter Colli, Suely Kazue Nagashi Marie, Lygia da Veiga Pereira, Maria Julia Manso Alves, Giuseppe Palmisano.

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
