## [Decision Letter · Decision Letter 0]

18 Jan 2024

Dear Dr. Palmisano,

Thank you very much for submitting your manuscript "ST8Sia2 polysialyltransferase protects against infection by Trypanosoma cruzi" for consideration at PLOS Neglected Tropical Diseases. As with all papers reviewed by the journal, your manuscript was reviewed by members of the editorial board and by several independent reviewers. In light of the reviews (below this email), we would like to invite the resubmission of a significantly-revised version that takes into account the reviewers' comments. 

We cannot make any decision about publication until we have seen the revised manuscript and your response to the reviewers' comments. Your revised manuscript is also likely to be sent to reviewers for further evaluation.

Sincerely,

Walderez O. Dutra, PhD.

Section Editor

Walderez Dutra

Section Editor

Reviewer's Responses to Questions

**Key Review Criteria Required for Acceptance?**

**Methods**

-Are the objectives of the study clearly articulated with a clear testable hypothesis stated?

-Is the study design appropriate to address the stated objectives?

-Is the population clearly described and appropriate for the hypothesis being tested?

-Is the sample size sufficient to ensure adequate power to address the hypothesis being tested?

-Were correct statistical analysis used to support conclusions?

-Are there concerns about ethical or regulatory requirements being met?

Reviewer #1: -Are the objectives of the study clearly articulated with a clear testable hypothesis stated? Yes

-Is the study design appropriate to address the stated objectives? Yes but see Summary and General Comments

-Is the population clearly described and appropriate for the hypothesis being tested? N/A

-Is the sample size sufficient to ensure adequate power to address the hypothesis being tested? Yes

-Were correct statistical analysis used to support conclusions? Yes

-Are there concerns about ethical or regulatory requirements being met? No

Reviewer #2: experiments well planned

**Results**

-Does the analysis presented match the analysis plan?

-Are the results clearly and completely presented?

-Are the figures (Tables, Images) of sufficient quality for clarity?

Reviewer #1: -Does the analysis presented match the analysis plan? Yes

-Are the results clearly and completely presented? Yes

-Are the figures (Tables, Images) of sufficient quality for clarity? Yes

Reviewer #2: excelent figures and images . Very clear graphs .

**Conclusions**

-Are the conclusions supported by the data presented?

-Are the limitations of analysis clearly described?

-Do the authors discuss how these data can be helpful to advance our understanding of the topic under study?

-Is public health relevance addressed?

Reviewer #1: -Are the conclusions supported by the data presented? Yes but see Summary and General Comments

-Are the limitations of analysis clearly described? 

-Do the authors discuss how these data can be helpful to advance our understanding of the topic under study?Yes

-Is public health relevance addressed? Yes

Reviewer #2: interesting and objectives conclusions. Maybe could speculate a little bit about the mechanism involved at the modulation.

**Editorial and Data Presentation Modifications?**

Reviewer #1: see Summary and General Comments

Reviewer #2: (No Response)

**Summary and General Comments**

Reviewer #1: This is an interesting paper that proposes that polysialic acid (synthesised by ST8Sia2) inhibits host cell invasion (and/or propagation?) by T. cruzi and that this is why T. cruzi infection of cells reduces ST8Sia2 expression. The data presented are well described but there are some missing controls, and the paper would benefit from more critical assessment of significance.

The authors should address the following points:

1. While the phenomena described are certainly noteworthy, it is difficult to understand an overall hypothesis as to why the phenomenon of T. cruzi infection of host cells causing downregulation of ST8Sia2 would influence an in vivo infection. Thus, the downregulation of ST8Sia2 in an individual cell would be post-invasion and unlikely(?) to influence cell invasion by more parasites, especially as it would take many hours for pre-existing polysialic acid to be turned over by the invaded host cell. The authors should help the reader understand how they think the T. cruzi invasion induced downregulation of ST8Sia2 expression could assist host infection and parasite survival. 

2. While the data presented using immunofluorescent microscopy data appear sound, they are difficult to assess. For example, in Fig 1A what % of host cells are infected and with what number (average and SD and median) of amastigotes? 

3. The reduction in ST8Sia2 mRNA is clear (Fig 1B) but the reduction in ST8Sia2 protein is slight (Fig 1B) (presumably because it has a long half-life – are there any data in the literature from global turnover studies?) bringing into question the reduction in polysialic acid as measured by Western (1D and G). 

4. The data are clearer in Fig 2 using SH-SY5Y cells instead of iPSc derived cells. However, SH-SY5Y cells are neuroblastoma cells (selected for their propensity to express polysialic acid, presumably(?)) but does infection of neurones occur in natural T. cruzi infections? What is the physiological basis for choosing this cell line?

5. Also in Fig. 2 B the variance (by eye) of signals to b-actin look wider than those on the plots. For example, the third uninfected sample b-actin signal is much stronger than the other two yet the ST8Sia2, PSA-NCAM and poly-Sia and NCAM1 signals look to be lower than the other two. All signals relative to b-actin in infected cells certainly look lower. However, if a large proportion of cells are infected (? – see point 1) why not perform this (for St8Sia2 and NCAM) by quantitative proteomics, an area of expertise in the author’s lab? 

6. Is the reduction in NCAM due to reduced protein stability or reduction in rate of protein synthesis? A pulse-chase experiment might resolve this issue. 

7. The data in Fig 3 with SH-SY5Y cells using anti-ST8Sia2 and anti-PSA immunofluorescence and EndoN-releases PSA measures are striking and seem to be mich more profound that the data in Fig 2 – why is that? Further, the uninfected micrograph seems to have a small number od cells with strong signal but most at background level closer to the signals seen in the infected cells micrograph (again what 5 cells are infected with how many amastigotes?) 

8. In Fig 3, where did the pre-existing PSA of the infected cells go to? 

9. The removal of cell surface PSA with EndoN makes the cells more easily invaded and/or allows that parasites to propagate better than in control cells. Since the only parameter reported is average number of amastigotes per cell it is impossible to judge which. Again, % infected cells and parasites/cell with SD are needed to assess the data. 

10. The data in Fig 5 are fascinating in principle but they lack important controls. We are asked to assume that CMP and ST8Sia2 siRNA do indeed reduce cell surface PSA (for some reason, that data is not presented using the techniques in the prior experiments) and that this, and this alone, is the reason for the increase in cell infection [Again, % infected cells and parasites/cell with SD are needed to assess the data, not just average amastigotes per cell]. Note: this reviewer is sceptical that exogenously added CMP would provide selective inhibition of Golgi enzyme ST8Sia2 (or even be taken up by cells at all). The authors should design experiments (perhaps more using EndoN?) where they can correlate extent of PSA reduction with infection efficiency.

Reviewer #2: The work of Barboza et al. contains important information regarding the modulation of trypanosoma cruzi of host machinery during infection and will be a great contribution in the area if published by PNTD.

The authors performed fine experiments associating infection, enzyme expression and acceptors of the transferred sugars, showing an inverse relationship between enzymatic activity and host polysialylation and infection load.To reach more general and conceptual conclusions, I will address a few points. 

1- It was very convincing to use two different lines of cells to see the effect of the infection and the phenotype of the cell, it would also be interesting to see two polar strains of t cruzi, one more and one less invasive to determine if there is a correlation with the presence of the enzymatic activity of ST8Sia2 and the products detected.5CN5A.

2- Decreased RNAI expression of ST8sia2 produced an increase in infectivity of T cruzi, would the opposite situation of overexpression of the enzyme produce a decrease in infection?

3-The authors do not work on or speculate what is happening in the modulation by the parasite. Parasitic extracts, fixed parasites, alteration of parasite surface, blockade of invasion by antibodies have impact on the enzymatic activity of ST8Sia2

PLOS authors have the option to publish the peer review history of their article (what does this mean?). If published, this will include your full peer review and any attached files.

Reviewer #1: No

Reviewer #2: No
---

## [Decision Letter · Decision Letter 1]

13 Aug 2024

Dear Dr. Palmisano,

We are pleased to inform you that your manuscript 'ST8Sia2 polysialyltransferase protects against infection by Trypanosoma cruzi' has been provisionally accepted for publication in PLOS Neglected Tropical Diseases.

Best regards,

Susan Madison-Antenucci, PhD

Section Editor

Walderez Dutra

Section Editor

Reviewer's Responses to Questions

**Key Review Criteria Required for Acceptance?**

**Methods**

-Are the objectives of the study clearly articulated with a clear testable hypothesis stated?

-Is the study design appropriate to address the stated objectives?

-Is the population clearly described and appropriate for the hypothesis being tested?

-Is the sample size sufficient to ensure adequate power to address the hypothesis being tested?

-Were correct statistical analysis used to support conclusions?

-Are there concerns about ethical or regulatory requirements being met?

Reviewer #2: this is ok

Reviewer #3: (No Response)

**Results**

-Does the analysis presented match the analysis plan?

-Are the results clearly and completely presented?

-Are the figures (Tables, Images) of sufficient quality for clarity?

Reviewer #2: this is ok

Reviewer #3: (No Response)

**Conclusions**

-Are the conclusions supported by the data presented?

-Are the limitations of analysis clearly described?

-Do the authors discuss how these data can be helpful to advance our understanding of the topic under study?

-Is public health relevance addressed?

Reviewer #2: this is ok

Reviewer #3: (No Response)

**Editorial and Data Presentation Modifications?**

Reviewer #2: this is ok

Reviewer #3: (No Response)

**Summary and General Comments**

Reviewer #2: the authors have improved the manuscript.

Reviewer #3: This is an interesting article with data that appropriately supports the manuscript's conclusions. Prior reviewer comments were appropriately addressed.

PLOS authors have the option to publish the peer review history of their article (what does this mean?). If published, this will include your full peer review and any attached files.

Reviewer #2: No

Reviewer #3: No

---

## [Editor Report · Acceptance letter]

3 Sep 2024

Dear Dr. Palmisano,

We are delighted to inform you that your manuscript, "ST8Sia2 polysialyltransferase protects against infection by Trypanosoma cruzi," has been formally accepted for publication in PLOS Neglected Tropical Diseases.

Best regards,

Shaden Kamhawi

co-Editor-in-Chief

Paul Brindley

co-Editor-in-Chief
